# The Q61H mutation decouples KRAS from upstream regulation and renders cancer cells resistant to SHP2 inhibitors

Teklab Gebregiworgis [1,10], Yoshihito Kano [2,3,9,10], Jonathan St-Germain[1], Nikolina Radulovich[1], Molly L. Udaskin[1], Ahmet Mentes[4], Richard Huang [2], Betty P. K. Poon[2,3], Wenguang He[3], Ivette Valencia-Sama[2,5], Claire M. Robinson[2,3], Melissa Huestis [2], Jinmin Miao [6], Jen Jen Yeh[7], Zhong-Yin Zhang [6], Meredith S. Irwin[2,5], Jeffrey E. Lee [2], Ming-Sound Tsao [1,2,8], Brian Raught[1,8], Christopher B. Marshall [1✉], Michael Ohh [2,3✉] & Mitsuhiko Ikura [1,8✉]

Cancer cells bearing distinct KRAS mutations exhibit variable sensitivity to SHP2 inhibitors (SHP2i). Here we show that cells harboring KRAS Q61H are uniquely resistant to SHP2i, and investigate the underlying mechanisms using biophysics, molecular dynamics, and cell-based approaches. Q61H mutation impairs intrinsic and GAP-mediated GTP hydrolysis, and impedes activation by SOS1, but does not alter tyrosyl phosphorylation. Wild-type and Q61H-mutant KRAS are both phosphorylated by Src on Tyr32 and Tyr64 and dephosphorylated by SHP2, however, SHP2i does not reduce ERK phosphorylation in KRAS Q61H cells. Phosphorylation of wild-type and Gly12-mutant KRAS, which are associated with sensitivity to SHP2i, confers resistance to regulation by GAP and GEF activities and impairs binding to RAF, whereas the near-complete GAP/GEF-resistance of KRAS Q61H remains unaltered, and high-affinity RAF interaction is retained. SHP2 can stimulate KRAS signaling by modulating GEF/GAP activities and dephosphorylating KRAS, processes that fail to regulate signaling of the Q61H mutant.

[1] Princess Margaret Cancer Centre, University Health Network, Toronto, Ontario, Canada M5G 1L7. [2] Department of Laboratory Medicine and Pathobiology, University of Toronto, Toronto, Ontario, Canada M5G 1M1. [3] Department of Biochemistry, University of Toronto, Toronto, Ontario, Canada M5G 1M1. [4] Black Diamond Therapeutics, New York, NY 10014, USA. [5] Department of Paediatrics and Cell Biology Program, Peter Gilgan Centre for Research and Learning, The Hospital for Sick Children, Toronto, Ontario, Canada M5G 0A4. [6] Department of Medicinal Chemistry and Molecular Pharmacology, Center for Cancer Research and Institute for Drug Discovery, Purdue University, West Lafayette, Indiana 47907, USA. [7] Lineberger Comprehensive Cancer Center and Departments of Surgery and Pharmacology, University of North Carolina, Chapel Hill, North Carolina 27599, USA. [8] Department of Medical Biophysics, University of Toronto, Toronto, Ontario, Canada M5G 1L7. [9] Present address: Department of Clinical Oncology, Graduate School of Medical and Dental Sciences, Tokyo Medical and Dental University, Tokyo, Japan. [10] These authors contributed equally: Teklab Gebregiworgis, Yoshihito Kano. ✉email: Chris.Marshall@uhnresearch.ca; michael.ohh@utoronto.ca; Mitsu.Ikura@uhnresearch.ca

RAS proteins are small GTPases that are regulated in a switch-like manner. They are turned "on" by binding a molecule of guanosine triphosphate (GTP) and turned "off" upon GTP hydrolysis. As the intrinsic rates of nucleotide exchange and GTP hydrolysis are slow, cells regulate these processes via proteins called guanine nucleotide-exchange factors (GEFs) and GTPase-activating proteins (GAPs). Activated RAS interacts with and activates effector proteins (e.g., RAF and phosphatidylinositol 3-kinase (PI3K) kinases), which drive cellular processes such as growth, proliferation, and metabolism. The RAS genes were among the first-discovered oncogenes and RAS mutation or amplification is associated with over 30% of all human cancers[1,2].

Of the three RAS isoforms (H-, K-, and N-RAS), 85% of oncogenic RAS mutations occur in KRAS[3]. More than 97% of these mutations occur at three hotspots (codons 12, 13, and 61) and multiple specific substitutions have been observed at each hotspot[4]. Codons 12 and 13 are the most frequently mutated sites in KRAS, whereas mutations in codon 61 are less common. By contrast, codon 61 is the most frequently mutated site in NRAS and HRAS, with Q61R/K being predominant[3]. Although all oncogenic mutations cause accumulation of the GTP-loaded form, each specific KRAS mutation exhibits some unique biochemical and structural properties[5–8]. Certain specific KRAS mutants have been associated with differential prognosis and therapeutic response of patients[9–13]. Thus, in-depth mutation-specific biochemical characterization of mutant RAS proteins is of paramount importance to understand the underlying mechanisms of pathogenesis, identify specific mutation-dependent therapeutic approaches, and, importantly, to identify patients who are likely to benefit from personalized or precision medicines[14,15].

SHP2 is a SH2 domain-containing protein tyrosine phosphatase encoded by the gene PTPN11, which has been known for over two decades to promote RAS-driven mitogen-activated protein kinase (MAPK) signaling, and activating PTPN11 mutations are one of the common causes of RASopathies, developmental syndromes defined by hyperactive RAS-MAPK signaling[16]. In recent years, there has been much excitement surrounding SHP2 inhibition as a potential therapeutic for KRAS-driven cancers. Our group and others have presented encouraging preclinical results that demonstrate the potential use of SHP2 inhibitors to impair growth and induce death of KRAS-driven cell lines, patient-derived organoids, and xenografts, alone or in combination with MEK, ERK, or ALK inhibitors[17–25]. We recently presented a model that demonstrates a direct catalytic role of SHP2 in reversing Src phosphorylation of KRAS. This model adds an additional layer to previously proposed roles of SHP2 in RAS-MAPK signaling and provides a molecular mechanism by which SHP2 inhibition prevents cell growth. Although SHP2 modulation of the KRAS GTPase cycle favors KRAS activation by promoting GEF[26,27] and restraining GAP activities[28,29], the phosphatase also acts directly on KRAS (Fig. 1a). Src kinase phosphorylates two tyrosine residues in the switch regions of KRAS, which impacts the GTPase cycle by disrupting regulation by GAPs and GEFs, as well as impairing binding to effector proteins[17,20,30]. Dephosphorylation by SHP2 releases KRAS from this "dark state," unleashing competent activated KRAS to restore signaling[17,20]. Interestingly, the nucleotide cycling properties of KRAS mutants have been proposed as predictors of sensitivity to SHP2 inhibition[18,20]. It has been reported, based on a limited number of cases, that cancer cells that harbor KRAS G13D or K/N-RASQ61X mutations are resistant to SHP2 inhibition[21,23,25,31]. However, the underlying mechanistic explanation for the differential sensitivity of distinct KRAS mutants towards SHP2 inhibition remains to be elucidated.

Here we show that pancreatic cancer cells that harbor KRAS Q61H mutation exhibit resistance to both catalytic and allosteric SHP2 inhibitors. KRAS Q61H, the most prevalent mutation occurring at codon 61 of KRAS[3], has been found in about 5% of pancreatic ductal adenocarcinoma (PDAC) patients[32,33] and has also been reported as a mechanism of acquired anti-epidermal growth factor receptor (EGFR) drug resistance in both lung and colorectal cancers[34–36]. Here we propose a detailed mechanistic explanation for the resistance of KRAS Q61H mutant cells to SHP2 inhibitors based on data from in vitro and cell-based biochemical analyses, as well as molecular dynamics (MD) simulations.

## Results

**PDAC cells harboring KRAS Q61H are resistant to SHP2 inhibitors.** We and others have shown that dampening KRAS signaling by targeting SHP2 may provide a tractable strategy for the treatment of cancers driven by the major (codon 12) oncogenic KRAS mutants, including non-small-cell lung cancer (NSCLC), gastroesophageal cancer, and PDAC[17–22]. However, recent data suggest that some KRAS mutants may not respond to SHP2 inhibition, raising important questions about the mechanisms of resistance. We observed that PDAC and PDAC patient-derived xenograft (PDX) cell lines harboring the KRAS Q61H mutation are less sensitive to the SHP2 catalytic inhibitor 11a-1[37], as well as the allosteric inhibitor SHP099[38], in comparison to cells harboring mutations at the hotspot codon 12 (G12D/V/C/R) (Fig. 1b and Supplementary Fig. 1). Spheroids prepared in ultra-low attachment, surface-coated round well-bottom microplates exhibited consistent results. Although the viability of CFPAC1 spheroids, which harbor KRAS G12V, was significantly decreased in the presence of SHP2i 11a-1, Hs766T spheroids, which harbor the KRAS Q61H mutant, were insensitive to the SHP2 inhibitor (Fig. 1c). Furthermore, this trend was also observed in PDAC patient-derived organoids grown in three-dimensional (3D) culture conditions. An organoid with KRAS Q61H mutation was less sensitive to SHP099 (GI$_{50}$ = 46.1 μM) compared to an organoid bearing a G12D KRAS mutation (GI$_{50}$ = 35.7 μM), whereas MiaPaCa-2, which is highly dependent on KRAS G12C was most sensitive (GI$_{50}$ = 22.1 μM) (Supplementary Fig. 2). Concordantly, phosphorylation of ERK (pERK) in SHP2i-sensitive codon 12-mutant PDAC and NSCLC, but not in Q61H-harboring resistant cells, was attenuated upon treatment with SHP2 inhibitors (Fig. 1d and Supplementary Fig. 1d). Furthermore, molecular ablation of SHP2 via CRISPR/Cas9-mediated knockout of PTPN11 in PDAC cell lines harboring KRAS G12V or G12D markedly reduced the level of pERK in response to epidermal growth factor (EGF) stimulation, whereas cells harboring KRAS Q61H exhibited no appreciable difference in pERK (Fig. 1e). SHP2 inhibitor treatment also attenuated the level of pERK in human embryonic kidney epithelial HEK293 cells transiently overexpressing wild-type (WT) KRAS or empty plasmid (i.e., mock-transfected cells expressing endogenous KRAS) in a dose-dependent manner, but failed to reduce pERK in cells overexpressing KRAS Q61H (Fig. 1f). Consistent with these results, HEK293 cells stably overexpressing GFP-KRAS Q61H exhibited resistance to SHP2 inhibitor treatment, whereas isogenic cells ectopically expressing GFP-KRAS WT or GFP alone were sensitive (Fig. 1g and Supplementary Fig. 3). These results suggest that some unique feature(s) of the KRAS Q61H mutant can render cancer cells independent of SHP2 function(s) and therefore resistant to pharmacologic (or molecular) inhibition of SHP2, unlike cells with WT-KRAS or the more common codon 12 mutations.

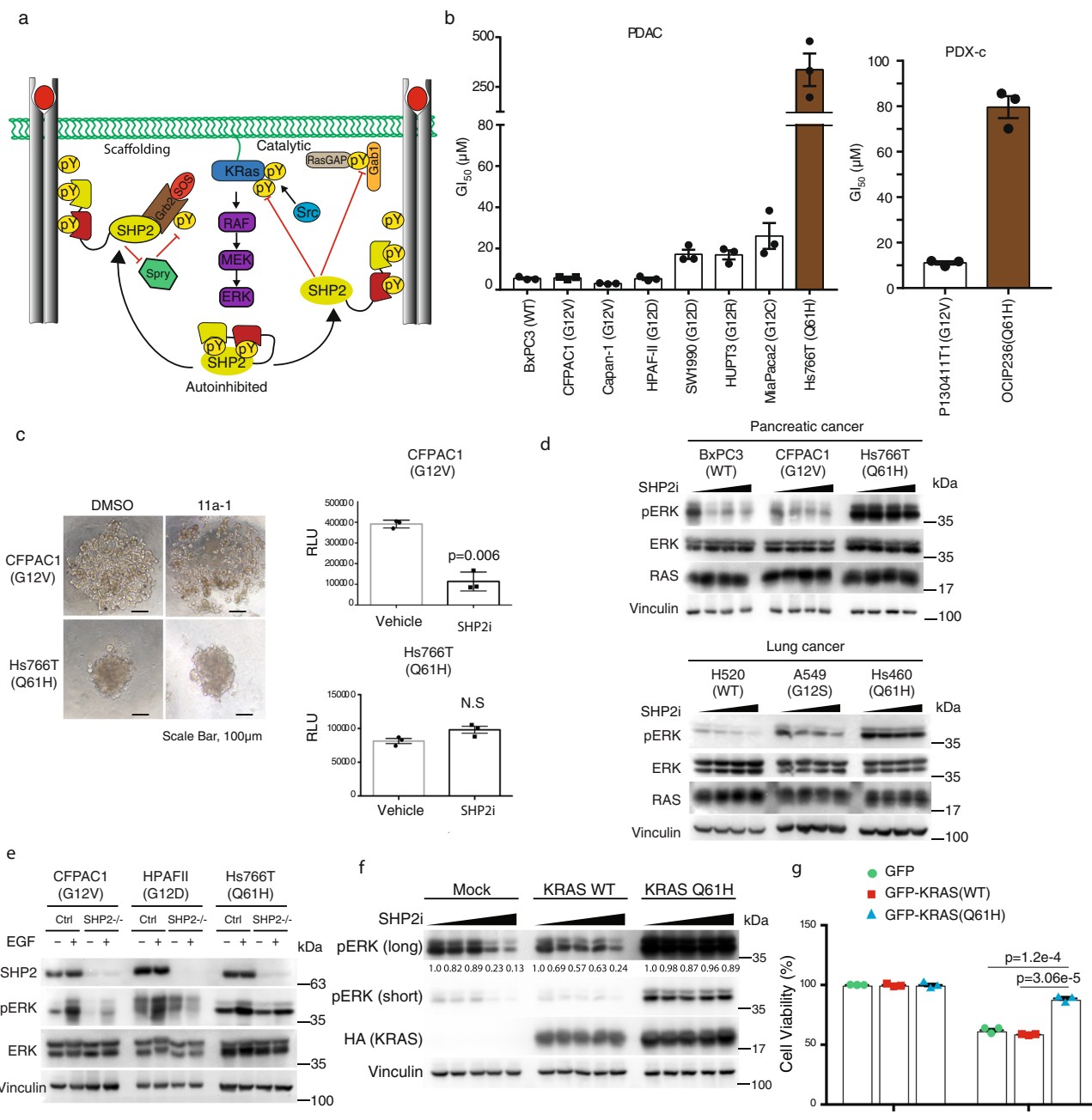

**Fig. 1 KRAS Q61H mutant cells exhibit insensitivity to pharmacologic inhibition of SHP2 (SHP2i). a** Schematic cartoon illustrating proposed catalytic and scaffolding roles of SHP2 in modulating KRAS signaling. SHP2 scaffolds the GRB2:SOS complex to activated receptor tyrosine kinases to promote RAS activation, whereas its phosphatase activity inhibits Sprouty's negative regulation of SOS (left) and dephosphorylates a pTyr docking site for p120RASGAP on GAB1, thereby reducing inactivation of KRAS (right). Top: SHP2 also dephosphorylates KRAS, release it from the "dark state" and restoring KRAS signaling. **b** PDAC cell lines (left) or PDX cells (right) were plated in 96-well plates in triplicate, treated with increasing concentrations of the SHP2 inhibitor 11a-1 for 48 h, then cell viability was determined using alamarBlue. $GI_{50}$ values were determined using GraphPad Prism 6.0. Data represent mean ± SEM of three independent experiments. **c** CFPAC1 or Hs766T cells were seeded at 1000 cells/well in 96-well spheroid microplates. Spheroids were cultured for 48 h and then exposed to vehicle (DMSO) or 50 µM 11a-1. Following 48 h incubation with 11a-1, a 3D cell viability assay was performed. Left: representative images obtained using a VWR Vista Vision inverted microscope. Scale bar, 100 µm. Right: data represent mean ± SEM. RLU (relative light units) of three independent experiments. To compare the 11a-1-treated and control group, *p*-value was generated using two-tailed Student's *t*-test. **d** PDAC or lung cancer cell lines were treated with the increasing concentrations of the SHP2 inhibitor SHP099 (0, 5, 10, 20 µM) for 2 h. Equal amounts of lysates were resolved on SDS-PAGE and immunoblotted with the indicated antibodies. **e** CRISPR/Cas9-mediated non-target (Ctrl) or SHP2−/− CFPAC1, HPAF-II, and Hs766T cells were serum-starved and treated with (+) or without (−) EGF (10 ng/ml). Equal amounts of lysates were immunoprecipitated and immunoblotted with the indicated antibodies. **f** HEK293 cells were transfected with mock, HA-KRAS WT, or HA-KRAS Q61H plasmids, as indicated. Cells were cultured for 48 h and then exposed to increasing concentrations of SHP099 (0, 2, 5, 10, 20 µM) for 2 h. Equal amounts of lysates were resolved on SDS-PAGE and immunoblotted with the indicated antibodies. Numbers indicate pERK levels normalized to vinculin based on densitometry. **g** HEK293 cells stably overexpressing GFP-KRAS WT, GFP-KRAS Q61H, or GFP with 11a-1 and cell viability was assessed at 48 h using alamarBlue. Data represent mean ± SEM of three independent experiments. The GFP and GFP-KRAS WT groups were independently compared with GFP-KRAS Q61H group using two-tailed Student's *t*-test. The respective *p*-values are indicated on the figure. **d**–**f** The blots are representative of three independent experiments.

**KRAS Q61H is insensitive to SOS1-mediated nucleotide exchange**. To investigate potential mechanisms for the SHP2 inhibitor resistance associated with Q61H, we first compared the nucleotide-exchange properties of six oncogenic KRAS mutants, as one of the established functions of SHP2 is to promote the GEF activity of Son of Sevenless (SOS1)[39]. To monitor the exchange reaction, a non-hydrolysable analog of GTP guanosine (guanosine 5'-O-[γ-thio] triphosphate, GTPγS) was added at tenfold molar excess to fully guanosine diphosphate (GDP)-loaded [15]N-KRAS samples (250 μM) and nucleotide exchange was observed using our real-time nuclear magnetic resonance (NMR) GTPase assay[40]. Initially, we determined the intrinsic nucleotide-exchange rate for WT and each of six KRAS mutants (Fig. 2a, b and Supplementary Table 1a). KRAS G13D exhibited the fastest intrinsic exchange, with a rate ~13-fold higher than WT, whereas KRAS Q61L was the next fastest (~4-fold faster than WT). The remaining mutants (G12V, G12D, G12C, and Q61H) exhibited intrinsic nucleotide-exchange rates similar to that of WT (Fig. 2a, b and Supplementary Table 1a). These trends are largely consistent with previously reported measurements that used fluorescently labeled nucleotide analogs[5], except the faster exchange we observed for Q61L was less pronounced in that study. The mutation at residue 61 (Q61H) did not appreciably accelerate the nucleotide exchange as measured by either method.

In cells, the activation of RAS is catalyzed by GEFs such as SOS1, which is recruited to receptor complexes upon growth factor stimulation. Although the intrinsic nucleotide exchange of common KRAS mutations was reported previously[5], the effects of specific KRAS mutations on GEF-mediated exchange have not been studied systematically. To address this gap, we investigated the impact of KRAS mutations on SOS1-assisted nucleotide-exchange kinetics mediated by recombinant SOS1 catalytic domain (residues 564–1049, hereafter referred to as SOS$^{cat}$). Interestingly, all oncogenic mutations tested reduced the sensitivity of KRAS to GEF activity. With the addition of SOS$^{cat}$ to KRAS at a molar ratio of 1:600, the WT protein exhibited >20-fold increase in exchange rate relative to the intrinsic rate, whereas G12V and G12D showed >5-fold increases. The G12C, G13D, and Q61L mutants exhibited strong impairment in their ability to be activated by SOS$^{cat}$ (two- to threefold enhancements). Notably, the Q61H mutant was completely non-responsive to the catalytic activity of SOS$^{cat}$, as its exchange rate was not increased above the intrinsic level (Fig. 2c, d, Supplementary Fig. 4a, and Supplementary Table 1a).

The crystal structure of the RAS:SOS complex shows that Gln61 is involved in a direct interaction with Thr935 of SOS (Fig. 2e, f), suggesting how its mutation to His may affect this interaction. To further investigate potential mechanism(s) underlying our experimental data, we performed MD simulations of WT-KRAS and the Q61H mutant in complex with the catalytic domain of SOS (residues 753–1046), based on crystal structures listed in Supplementary Table 3. Previous structural studies showed that RAS Switch II forms an extensive network of interactions with SOS to stabilize the KRAS:SOS complex[41], and that mutation of several Switch II residues hinder interaction with SOS and/or catalysis of nucleotide exchange[42,43]. Our MD simulations reveal that when complexed with SOS$_{cat}$, the P-loop, switch II, and alpha3 regions are more dynamic in WT-KRAS vs. the Q61H mutant, on the basis of root mean square deviation and root mean square fluctuation measurements of C-α atoms (Supplementary Fig. 5a–d). Fewer interactions were observed between KRAS residues, including Ala59 and Tyr71, with SOS in the Q61H mutant vs. WT (Supplementary Fig. 6a, b). The substitution of the His sidechain for Gln61 severely disrupted the Gln61:Thr935 interaction (Fig. 2e, f). Hydrophobic interactions between RAS and SOS also make important contributions to

stabilizing the RAS:SOS complex. For example, RAS Tyr64 becomes buried in a hydrophobic patch of SOS formed by Ile825 and Phe929[41], and mutation of Tyr64 to Ala was shown to reduce the binding affinity of HRAS to SOS by at least 50%[42]. Consistently, the sidechain of Tyr64 of WT-KRAS remained clamped between SOS Ile825 and Phe929 during our entire MD simulation, whereas Tyr64 of the Q61H mutant dissociated from this hydrophobic groove after 690 ns (Supplementary Fig. 6c). These simulations strongly suggest that the Q61H mutation reduces the stability of the SOS:RAS complex, consistent with our observation that the mutation disrupts SOS-mediated nucleotide exchange (Fig. 2a–d and Supplementary Fig. 4a).

**KRAS Q61H is insensitive to the GAP activity of RASA1**. To characterize the other side of the KRAS Q61H GTPase cycle, we measured its intrinsic GTP hydrolysis and sensitivity to the GAP activity of RASA1. Consistent with the known role of Gln61 in hydrolysis, our results demonstrate that the Q61H mutation decreases the intrinsic GTP hydrolysis rate by about threefold compared to the WT and, importantly, the mutant was completely resistant to stimulation of GTP hydrolysis by the GAP domain of RASA1 (1:3000 molar ratio), whereas the hydrolysis rate of WT-KRAS was enhanced by over 300% in the same condition (Fig. 2g, Supplementary Fig. 4b, and Supplementary Table 1b).

To investigate potential mechanisms underlying our experimental observation of the exceptional GAP resistance of KRAS Q61H at the atomic level, we conducted all-atom MD simulations of KRAS Q61H in complex with RasGAP using crystal structures listed in Supplementary Table 3. The simulation revealed residues that may stabilize the protein–protein interaction through a network of hydrogen bonds, salt bridges, and hydrophobic interactions between RAS and the GAP domain. Unlike the mutant-KRAS:SOS complex, the mutant-KRAS:RasGAP complex remained intact, stabilized by several long-lived salt bridges, hydrogen bonds, and hydrophobic interactions (details in Supplementary Fig. 7a–h). These data suggest that the Q61H mutation does not disrupt the formation of the KRAS:RasGAP complex. Rather, KRAS Q61H was predicted to form a mutant-specific interaction with the extra domain of RasGAP (Supplementary Fig. 8a–c), which could potentially enhance stability of the complex. Thus, we focused on the local structure of the catalytic regions in the simulations. Previously, it has been proposed that a hydrogen bond between the "arginine finger" (Arg789) and Gln61 promotes activation of the catalytic water molecule[44,45]. In our simulation of the Q61H mutant, the backbone and sidechain of His61 were pointed away from the catalytic site during most of the simulation and did not interact with the arginine finger (Fig. 2h and Supplementary Fig. 9). Taken together, our simulation strongly suggests that the Q61H mutation alters the local structure of the catalytic site and directly impairs the previously proposed roles of Gln61 in stabilizing the catalytic residues and/or acting as a general base to extract a proton from the catalytic water molecule[41,46–48].

In summary, these findings demonstrate that the KRAS Q61H GTPase cycle is severely decoupled from regulation by the GEF and GAP activities of SOS1 and RASA1, whereas the properties of its intrinsic GTPase cycle would lead to the accumulation of the GTP-loaded form.

**KRAS Q61H is phosphorylated by Src and dephosphorylated by SHP2**. Previously, we described a molecular mechanism by which SHP2 inhibition can dampen KRAS signaling[17]. In our model, Src phosphorylation of the switch regions of KRAS stalls its GTPase cycle and impairs its binding to effector proteins,

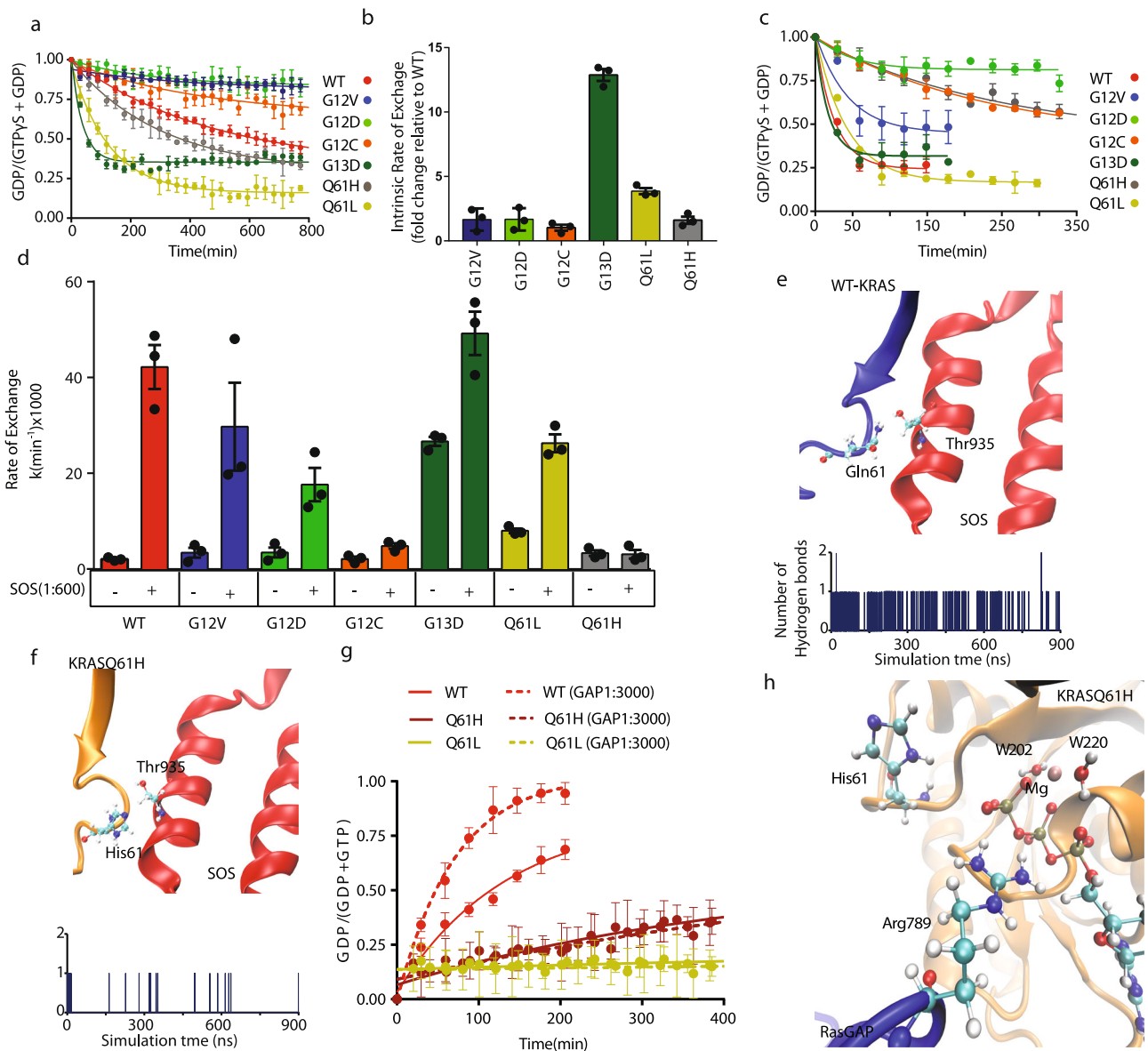

**Fig. 2 KRAS Q61H is insensitive to GEF and GAP activities. a** Intrinsic nucleotide-exchange curves of wild-type (WT) and mutant KRAS obtained from real-time NMR experiments. A tenfold molar excess of GTPγS was added to GDP-loaded KRAS at $t = 0$ and the GDP-bound fraction remaining was monitored over time by NMR. The data were fit to one-phase exponential decay functions. The nucleotide-exchange curves do not go to zero, because the nucleotide exchange equilibrates at a certain ratio, which is a function of the relative affinity of GDP:GTPγS. **b** Histogram of intrinsic exchange rates of each mutant determined in **a**, presented as fold changes for the mutants relative to wild-type KRAS. The error bars indicate the SD of the fold changes. **c** GEF-assisted nucleotide-exchange curves in the presence of SOS^cat (1 : 600 molar ratio SOS : KRAS), measured using real-time NMR. Data were fit as in **a**. **d** Histogram representing the mean intrinsic and SOS^cat (1 : 600) assisted exchange rates for wild-type KRAS and mutants obtained from three independent fitting curves for each mutation. Three pair of peak intensities for each group were used to generate the fitting curves. The error bars represent SEM of the three exchange rates. **e** A snapshot ribbon diagram of SOS1 (red) in complex with WT-KRAS (blue) (top panel). Time evolution of hydrogen bonds formed between the Thr935 of SOS and the sidechain of KRAS Q61 (bottom panel). **f** A snapshot ribbon diagram of SOS1 (red) in complex with KRAS Q61H (orange) (top panel). Time evolution of hydrogen bonds formed between the Thr935 of SOS and the sidechain of KRAS His61. **g** Intrinsic and GAP-assisted GTP hydrolysis curves for the wild type and Q61H mutant. The GAP domain of RASA1 was added at a ratio of 1 : 3000 to KRAS-GTP (**a**, **c**, **e**). **h** A ribbon diagram of RasGAP (blue) and KRAS Q61H (orange) catalytic site. Dots represent the fraction of GDP-loaded KRAS based on the intensities of GDP- and GTP-specific peaks ($I_{GDP}/I_{GDP} + I_{GTPγS}$). **a**, **b**, **g** Error bars represent SEM of the three independent pairs of peak intensities used for the calculation.

whereas SHP2 dephosphorylates these sites to restore KRAS signaling to MAPK[17]. Thus, SHP2 inhibitors prevent the reactivation of KRAS by SHP2. To dissect the molecular mechanisms of resistance to SHP2 inhibitors in cells that harbor KRAS Q61H mutation, we initially examined whether the mutation directly altered the phosphorylation profile of KRAS. HEK293 cells transfected with plasmids encoding Src together with HA-KRAS

WT, G12D, G12V, or Q61H were immunoprecipitated using anti-hemagglutinin (HA) antibody and probed with an antibody recognizing phosphorylated tyrosine (pan pTyr). Less phosphorylated KRAS (pKRAS) was detected for all mutants compared to WT; however, the amount of phosphorylated Q61H was not less than the other mutants (Fig. 3a), suggesting that SHP2 resistance is not due to a lack of phosphorylation of this mutant.

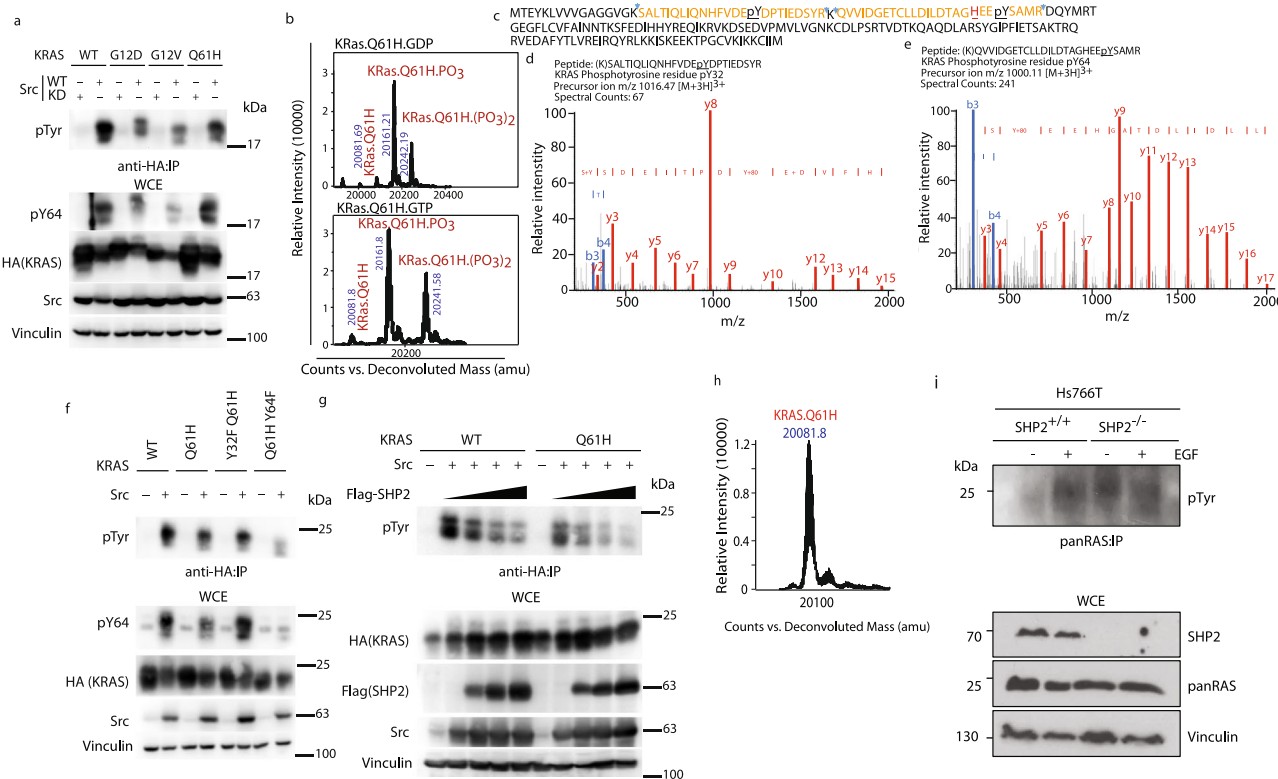

**Fig. 3 KRAS Q61H is phosphorylated by Src and dephosphorylated by SHP2. a** HEK293 cells were transfected with HA-tagged KRAS WT, G12D, G12V, or Q61H and either kinase-dead (KD) or active (WT) Src. KRAS was immunoprecipitated with anti-HA antibody, resolved on SDS-PAGE, and immunoblotted with anti-pTyr. Equal amounts of lysates were resolved on SDS-PAGE and immunoblotted with an antibody recognizing phosphorylated KRAS Y64, as well as the antibodies indicated to confirm expression and loading. **b** KRAS Q61H loaded with GDP or a non-hydrolysable GTP-analog (GMPPNP) was incubated with Src (1 : 250 Src to KRAS ratio) and ATP, and the reaction products were analyzed with mass spectrometry. The predicted mass of $^{15}$N-KRAS Q61H (residues 1–173 C118S) is 20084.36. Peaks corresponding to unmodified KRAS Q61H, as well as KRAS Q61H phosphorylated at one (+80 Da) and two sites (+160 Da) are labeled accordingly. **c** KRAS4B Q61H sequence indicating tryptic peptides in which phosphotyrosine was detected (orange letters with blue arrows indicating trypsin cleavage sites). **d, e** LC-MS analyses of phosphorylated tyrosine residues in KRAS Q61H. Two phosphorylated tyrosine residues from the KRAS Q61H protein sequence were identified from in vitro kinase reaction samples that were reduced/alkylated, trypsin-digested, and analyzed by LC-MS. MS/MS spectra matched KRAS tryptic peptides containing phosphorylated **d** Y32 and **e** Y64. **f** HEK293 cells were transfected with HA-KRAS WT, Q61H, Y32F/Q61H, or Q61H/Y64F mutants, with or without co-transfection of Src. HA-KRAS was immunoprecipitated with anti-HA and probed with anti-pTyr antibody. The whole-cell extracts were resolved on SDS-PAGE and immunoblotted with an antibody recognizing KRAS pTyr64, as well as those indicated, to confirm expression and loading. **g** HEK293 cells were transfected with HA-KRAS WT or Q61H increasing amounts of SHP2, with or without co-transfection of Src. HA-KRAS was immunoprecipitated with HA antibody and blotted with pTyr antibody. The whole-cell extracts (WCEs) were blotted with the antibodies indicated to confirm expression and loading. All immunoblot data are representative of at least three independent experiments. **h** Src-phosphorylated KRAS Q61H was incubated with SHP2 for 2 h and the total mass was analyzed using LC/MS as explained in "Methods." The predicted mass of unmodified $^{15}$N-KRAS Q61H (residues 1–173 C118S) is 20084.36. **i** CRISPR/Cas9-mediated non-target (SHP2+/+) or SHP2−/− Hs766T cells were serum-starved for 20 h and and treated with (200 ng/ml) (+) or without (−) EGF for 1.5 min. One milligram of lysates was immunoprecipitated and immunoblotted with the indicated antibodies. **a, f, g, i** The blots are representative of three independent experiments.

The differential phosphorylation of WT-KRAS vs. mutants may be related to their states of activation in the cell. To investigate the effect of activation on phosphorylation, we performed in vitro phosphorylation by incubating recombinant KRAS Q61H preloaded with GDP or the GTP-analog Gpp(NH)p (Guanosine 5′-[β,γ-imido]triphosphate trisodium salt hydrate) with c-Src (1 mol of Src to 125 mol of KRAS), 2 mM of ATP, and phosphatase inhibitors. The reactions were analyzed by mass spectrometry (MS) to investigate the extent of the phosphorylation. The GDP- and GTP-analog-loaded KRAS Q61H samples were both phosphorylated at one to two sites with similar distributions of mono- and diphosphorylated forms (Fig. 3b), which is similar to our previous observations for WT-KRAS[17]. This indicates that phosphorylation is not dependent on the nucleotide bound.

Liquid chromatography-tandem MS (LC-MS/MS) analysis of trypsin-digested KRAS Q61H identified Tyr32 and Tyr64 as the phosphorylation sites in KRAS Q61H, with Tyr64 representing the most frequently observed phosphorylated site (Fig. 3c–e). This in vitro observation was confirmed using a newly developed phospho-(p)Y64-specific anti-RAS antibody. The specificity of this antibody was verified using purified KRAS in the presence of active Src and pY64 RAS signal was strongly competed away by excess pY64 peptide, but not the non-phosphorylated form of the same peptide (Supplementary Fig. 10a). We further validated the antibody by phosphorylating purified WT-KRAS and mutants with phenylalanine substitutions at Tyr32, 40 or 64, as well as a double mutant (32/64) with Src. The antibody generated robust signals in WT pKRAS, Y32F, and Y40F, but not in Y64F or the Y32/64F double mutant, thereby demonstrating the specificity of the antibody for pY64 (Supplementary Fig. 10b). This pY64 antibody detected higher levels of Y64 phosphorylation in the Q61H mutant compared to G12D or G12V (Fig. 3a). Moreover, the pan-pTyr antibody revealed that KRAS Q61H/Y64F double

mutant had a marked diminution in Src-mediated phosphorylation, while KRAS Y32F/Q61H retained relatively robust tyrosyl phosphorylation (Fig. 3f). These results suggest that Y64 is the primary site of Src phosphorylation on KRAS Q61H, similar to WT-KRAS.

We then investigated the possibility that the Src-phosphorylated pKRAS Q61H may be resistant to SHP2 dephosphorylation. In HEK293 cells, ectopic expression of Flag-WT-SHP2, but not a catalytically dead SHP2 mutant (C459S), decreased the levels of Src-phosphorylated HA-pKRAS Q61H, similar to WT pKRAS (Fig. 3g and Supplementary Fig. 11). Furthermore, incubation of pKRAS Q61H with recombinant SHP2 in vitro reversed phosphorylation (Fig. 3h), indicating that the Q61H mutation does not interfere with SHP2 dephosphorylation of KRAS. Taken together, the Q61H mutation does not directly impact tyrosyl phosphorylation or dephosphorylation of KRAS, even though it is proximal to the major phosphorylation site (Y64). The observation that the KRAS mutants are less phosphorylated than WT-KRAS in vivo may be related to the accessibility of these tyrosine residues to Src, i.e., GTP-loaded mutants engaging effector proteins would be partly protected from Src kinase activity. Indeed, addition of BRAF RAS-binding domain (RBD) to GTP-loaded KRAS in vitro blocked the phosphorylation of KRAS by the Src kinase domain (Supplementary Fig. 12).

Finally, we sought to investigate phosphorylation of endogenous KRAS in a human pancreatic cancer cell line with KRAS Q61H mutation (Hs766T). First, we rigorously evaluated the available antibodies to pull down or detect KRAS. By blotting Hs766T lysates alongside those of HEK293T cells overexpressing HA-tagged K-, H-, and N-RAS using RAS isoform-specific panRAS and anti-HA antibodies, we determined that the Hs766T cell line expresses K-RAS much more highly than H- or N-RAS (Supplementary Fig. 13). Although the anti-HA blot demonstrates that all three tagged isoforms expressed well, H- and N-RAS were better detected by their respective isoform-specific antibodies than was KRAS (antibody OP24). Despite the differential sensitivities of the three isoform-specific antibodies, the KRAS band was still the main isoform detected in Hs766T. Having established that KRAS is the major RAS isoform in these cells, we used the superior anti-panRAS antibody for immunoprecipitation of KRAS and visualized tyrosyl phosphorylation using a pan-pTyr antibody (4G10). The result shows that tyrosyl phosphorylation was induced by EGF treatment (Fig. 3i). Moreover, in an isogenic pancreatic cancer cell line with CRISPR-Cas9-mediated knockout of SHP2, tyrosyl phosphorylation of endogenous KRAS Q61H was observed even in the absence of EGF treatment, which is consistent with the notion that SHP2 is involved in dephosphorylating endogenous KRAS (Fig. 3i).

**Src phosphorylation promotes the intrinsic exchange rate of KRAS Q61H.** As the resistance to SHP2i associated with the KRAS Q61H mutation does not appear to be related to differential phosphorylation, we examined the impacts of phosphorylation on the biochemical properties of the mutant. To understand the role of phosphorylation in regulating the GTPase cycle of KRAS Q61H, we purified pKRAS Q61H mutant (pKRAS Q61H) and characterized its GTPase cycle using real-time NMR. Phosphorylation increased the intrinsic exchange rate of KRAS Q61H by more than >5-fold (Fig. 4a, b and Supplementary Table 3); however, the addition of SOS$^{cat}$ (1 : 600 ratio) did not stimulate the exchange reaction further. Phosphorylation slightly decreased the intrinsic GTP hydrolysis rate of pKRAS Q61H and the addition of GAP had no impact on hydrolysis, similar to the non-pKRAS Q61H (Fig. 4c, d and Supplementary Table 2b). Thus, as described above, the Q61H mutation uncouples KRAS

from regulation of its GTPase cycle by GAPs and GEFs, and whereas phosphorylation reduces the sensitivity of WT-KRAS to these regulators, KRAS Q61H is unaffected, remaining insensitive. The properties of the intrinsic GTPase cycle of KRAS Q61H lead to accumulation of the GTP-loaded state through impaired hydrolysis and this is accentuated by phosphorylation, which further suppresses intrinsic GTP hydrolysis, while also accelerating intrinsic nucleotide exchange. Notably, the Q61H mutant is distinct in that its nucleotide exchange is enhanced only by phosphorylation, but not the GEF activity of SOS.

To study the impact of mutation and phosphorylation on the KRAS structure, we compared the $^1$H-$^{15}$N heteronuclear single quantum coherence (HSQC) NMR spectra of unphosphorylated and phosphorylated samples of WT and Q61H KRAS. Analysis of an overlay of four spectra (the phosphorylated and unmodified spectra of WT and Q61H) reveals similar patterns of chemical-shift changes associated with the Q61H mutation and phosphorylation (Fig. 4e). A number of these chemical-shift changes associated with mutation and phosphorylation affect peaks from residues in switch I and II, suggesting structural perturbations of this region, which is involved in coordinating the nucleotide and docking of GAPs and GEFs, likely contribute to the biochemical outcomes. Switch I is also the binding site for effector proteins; thus, we examined how mutation and phosphorylation affect KRAS interaction with an RBD.

**KRAS Q61H is resistant to phosphorylation-dependent regulation of MAPK signaling.** Our previous study showed that the signaling of WT or common mutants of H/NRAS such as G12V and 12D can be suppressed by promoting their tyrosyl phosphorylation via ectopic Src expression or molecular inhibition of SHP2[30,49]. Similarly, we observed that in addition to decoupling KRAS from regulation of the GTPase cycle, tyrosyl phosphorylation of KRAS WT or G12V mutant was associated with reduced binding affinity to RAF[17]. In contrast to these cases, KRAS Q61H is already decoupled from regulation by GEFs and GAPs; however, Src phosphorylation of Q61H promotes intrinsic nucleotide exchange and leads to the accumulation of the GTP-loaded form, which should contribute to downstream signaling. Thus, we investigated the impact of KRAS Q61H phosphorylation on BRAF-RBD binding. First, we phosphorylated KRAS Q61H and separated the mono- and diphosphorylated fractions using ion-exchange chromatography (Supplementary Fig. 14a), to perform quantitative in vitro binding assays. We confirmed the phosphorylation status of the samples using MS (Supplementary Fig. 14b), then measured the affinities for RBD of the unmodified, mono-, and diphosphorylated KRAS Q61H proteins using a biolayer interferometry biosensor (Octet) (Supplementary Fig. 14c). Diphosphorylated KRAS Q61H samples exhibited ~4-fold weaker binding than the unmodified sample, whereas the same modification reduced WT-KRAS binding to the RBD by more than 15-fold (Fig. 5a and Supplementary Fig. 14c, d). Further, RBD pulldown experiments of KRAS expressed in HEK293 cells produced consistent results. In cells co-transfected with KRAS G12V and increasing amounts of Src, the amount of KRAS pulled down by immobilized RAF1-RBD decreased with increasing Src expression; however, the KRAS Q61H mutant did not exhibit such a marked impairment (Fig. 5b).

To better understand the impact of phosphorylation of KRAS on its interaction with RAF-RBD at the atomic level, we performed all-atom MD simulations of four KRAS-RAF1 complexes (KRAS-RAF1, pKRAS-RAF1, KRAS Q61H-RAF1, and pKRAS Q61H-RAF1) (Supplementary Table 3 and Supplementary Fig. 15). The unmodified KRAS-RAF1 and the KRAS Q61H-RAF1 systems share common KRAS-RAF contact points that persist throughout our simulation[50]. Several hydrogen bonds

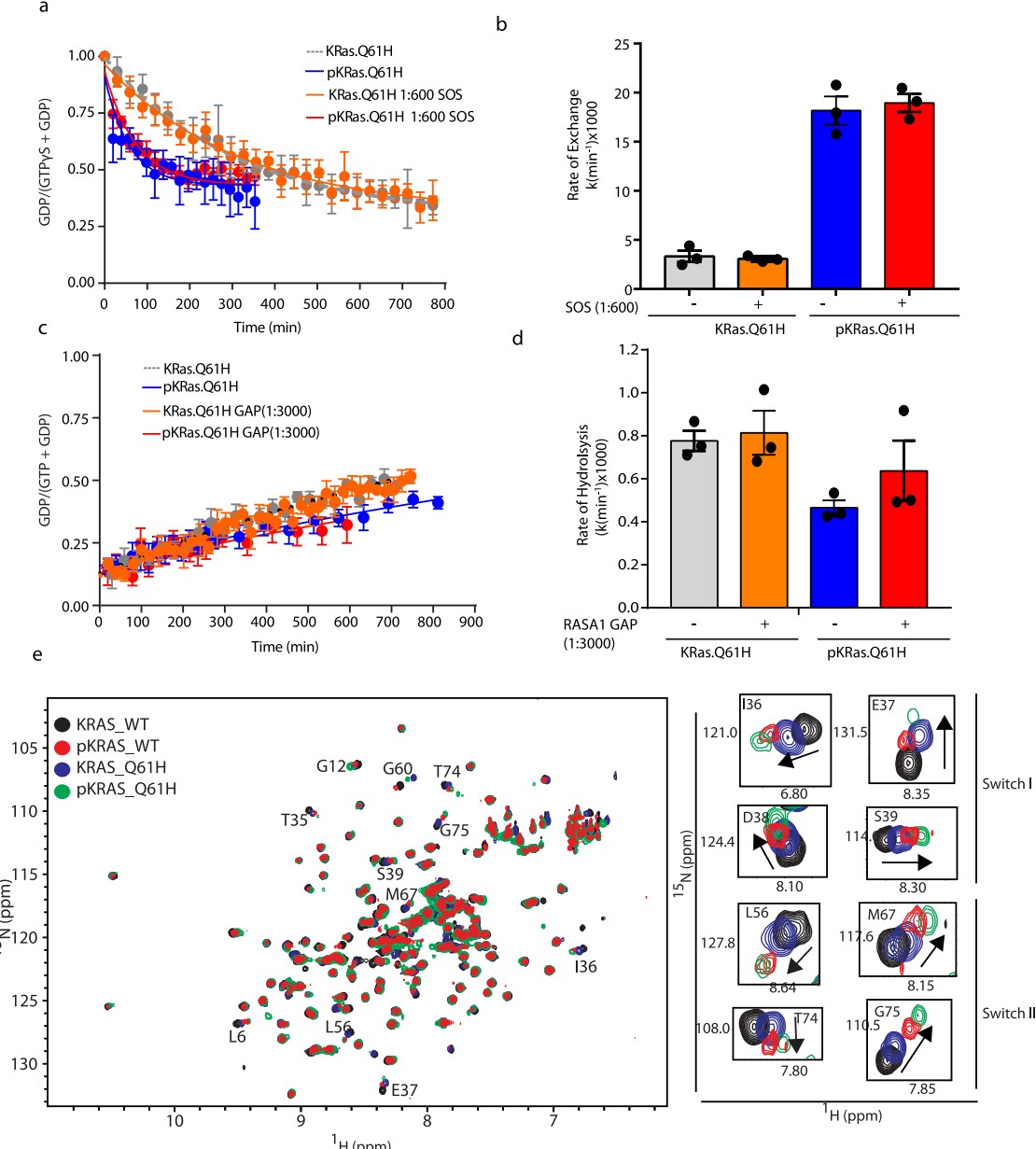

**Fig. 4 Phosphorylation enhances the intrinsic exchange of KRAS Q61H. a** Intrinsic and SOS$^{cat}$ (1 : 600 molar ratio) assisted KRAS nucleotide-exchange curves of unmodified and phosphorylated KRAS Q61H. **b** Nucleotide-exchange rates derived by fitting the curves in **a**. **c** Intrinsic and RASA1 GAP (1 : 3000 molar ratio)-catalyzed KRAS-GTP hydrolysis curves of unmodified and phosphorylated KRAS Q61H. **d** GTP hydrolysis rates derived by fitting the curves in **c**. **e** Overlaid $^1$H-$^{15}$N HSQC NMR spectra of wild-type KRAS (WT) and the Q61H mutant, as well as their phosphorylated forms (pWT and pQ61H). Peaks for selected Switch I and II residues are expanded on the right. **a–e** Phosphorylated KRAS samples were generated by incubating $^{15}$N-KRAS with Src (1 : 250 ratio) and ATP followed by passage through a Superdex 75 10/300 column (GE Healthcare). Nucleotide exchange and hydrolysis rates were measured using real-time NMR, starting with fully GDP- and GTP-loaded KRAS Q61H samples, respectively. **a**, **c** Dots represent the fraction of KRAS that is GDP loaded ($I_{GDP}/I_{GDP} + I_{GTP\gamma S}$) based on the intensities of peaks from three residues. The line was obtained by fitting a one-phase exponential decay curve. Error bars indicate the SEM of the rate calculated from three residues peak intensities. **b**, **d** Histograms indicate the mean exchange or hydrolysis rates calculated from intensities of three residues. The error bars represent SEM of the respective rates.

between KRAS switch I and RAF were observed in our MD simulation (Supplementary Table 4), whereas Tyr32 and Tyr64 were not interacting with the RBD. In the unmodified WT-KRAS-RAF1 simulation, Tyr32 was directed away from the RBD interaction site and formed an interaction with the γ-phosphate of GTP stabilized by a long-resident Na$^+$ ion (Supplementary Fig. 16a, b). Tyr64 is positioned far from the KRAS-RAF1 contact region. Upon phosphorylation of WT-KRAS, the model predicted that pTyr32 and pTyr64 forms new interactions with RAF1 (with

Arg73 and Arg59/Thr54, respectively), whereas the modification disrupted the interaction between Glu37 of WT-KRAS and Arg59 of RAF1 (Supplementary Fig. 17 and Supplementary Table 4). In our simulation, the distinctive characteristics of the pKRAS-RAF1 complex can be attributed to the dynamics of the pKRAS switch regions. The simulations revealed that phosphorylation increases the mobility of switch I and II of WT-KRAS, whereas the KRAS Q61H protein becomes stiffer upon phosphorylation (Fig. 5c). Our result is consistent with a previous finding that tyrosyl

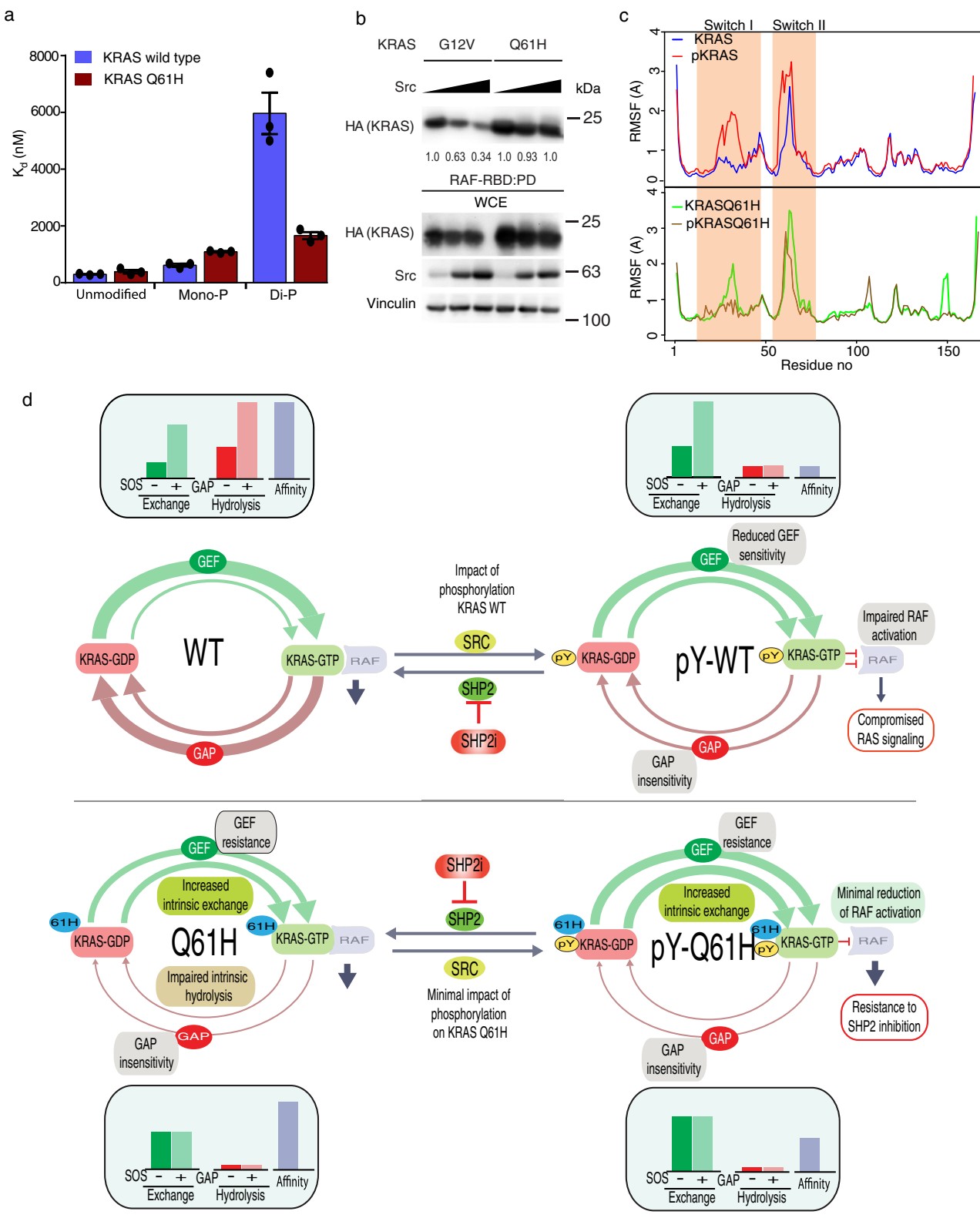

phosphorylation alters the conformational dynamics of the KRAS protein[51]. Furthermore, the phosphorylation of WT-KRAS caused a substantial allosteric effects on the dynamics of the bound RAF1-RBD.

These experimental and MD simulation results indicate that Src phosphorylation of KRAS at Y32 and Y64 positions has less impact on RAF-RBD binding of the Q61H mutant compared to WT protein and the G12V mutant, which is consistent with the failure of SHP2 inhibition to reduce ERK phosphorylation in cells with endogenous or exogenous KRAS Q61H.

## Discussion

Our results show that cells with KRAS Q61H mutation are resistant to SHP2 inhibition as measured by ERK

**Fig. 5 KRAS Q61H is resistant to regulation of RAF-RBD binding by tyrosyl phosphorylation. a** Histogram representing the dissociation constant ($K_d$) for the binding of unmodified, singly or doubly phosphorylated wild-type or Q61H mutant KRAS to BRAF-RBD. The phosphorylated forms were separated using ion-exchange chromatography and binding to immobilized RBD was measured using BLI. The error bars indicate SEM of three independent measurements. **b** HEK293 cells were transfected with G12V or Q61H HA-KRAS together with increasing amounts of Src. Cells were lysed and activated HA-KRAS was pulled down with RAF1:RBD-conjugated beads and immunoblotted with anti-HA. The whole-cell extract (WCE) was resolved by SDS-PAGE and probed with the indicated antibodies to confirm expression and loading. Numbers indicate normalized ratios of activated KRAS to total KRAS based on densitometry. The immunoblottings are representative of at least three independent experiments. **c** A line graph that shows root mean square fluctuation (RMSF) of WT-KRAS (top, modified-red, unmodified-blue) and KRAS Q61 (bottom, modified-brown, unmodified-green). The X-axis shows the residue id for the proteins and the Y-axis is for fluctuation distance in Å. **d** Model summarizing the impact of Q61H mutation and tyrosyl phosphorylation on the KRAS GTPase cycle and RBD binding. Line thicknesses represent relative rates of each step in the GTPase cycle. The upper panels (from left to right) show the impact of phosphorylation on the GTPase cycle of KRAS WT: reduced sensitivity to the activity of GAPs and GEFs, and impaired binding to RAF. The left panels (from top to bottom) show the impact of the Q61H mutation on the GTPase cycle of KRAS: reduced sensitivity to the activity of GAPs and GEFs with little impact on RAF. The lower panels (from left to right) represent how phosphorylation of the KRAS Q61H mutant has little additional impact on its GTPase cycle.

phosphorylation and cell viability (Fig. 1). The finding is consistent with recent reports[20,21] showing the sensitivity of cells with specific KRAS mutations to SHP2 inhibitors is associated with the properties of the KRAS GTPase cycle of each mutant. Cells with codon 12 mutations exhibit higher sensitivity than cells with codon 13 and 61 mutations[25,52], but the underlying mechanistic explanation of this phenomenon has remained unclear[21,53].

SHP2 is an SH2 domain-containing phosphatase that activates the MAPK pathway. Earlier investigations show that both catalytic and scaffolding roles of SHP2 regulate the RAS GTPase cycle (Fig. 1a). SHP2 acts as a scaffold to recruit the GRB2:SOS complex to the plasma membrane to promote RAS activation[26,27], whereas its phosphatase activity (i) inhibits Sprouty's negative regulation of SOS to maintain GEF function[54], (ii) eliminates a docking site for p120RASGAP on EGFR and GAB1[28,29] to prevent rapid inactivation of KRAS-GTP, and (iii) directly dephosphorylates Src-phosphorylated RAS to restore the canonical GTPase cycle and RAF-binding affinity[17]. Distinct biochemical properties of specific KRAS mutants likely determine whether they confer resistance or exhibit sensitivity to SHP2 inhibitors, based on whether their activation and signaling depend on these SHP2 functions.

To characterize and compare the properties of the GTPase cycle of KRAS Q61H with the other common mutants, we first profiled the intrinsic and SOS^cat catalyzed nucleotide exchange of six KRAS mutants. Of the samples we tested, KRAS Q61H was the only mutant that was completely insensitive to the GEF activity of SOS^cat, whereas nucleotide exchange of the other mutants was accelerated by SOS^cat, albeit less efficiently than WT-KRAS (Fig. 2 and Supplementary Fig. 4a). In addition, the Q61H mutation reduced the intrinsic GTPase activity by about threefold compared to the WT and rendered KRAS Q61H resistant to RASA1 GAP stimulation. These properties suggest that the KRAS Q61H mutant is not dependent on SOS or on the upstream Receptor Tyrosine Kinases (RTKs) signaling that stimulates SOS and is thus decoupled from the aforementioned mechanisms by which SHP2 function promotes KRAS signaling through SOS^cat or by restricting RASA1 GAP activity[26–29]. As the catalytic domains of all RAS GAPs and GEFs are structurally and mechanistically similar[55], we anticipate the Q61H mutation would similarly confer resistance to the activities of other RAS GAPs and GEFs. In parallel to these findings, previous studies have reported frequent emergence of KRAS Q61H mutation as a molecular mechanism of acquired resistance to EGFR inhibition[34,35]. Interestingly, cell-free DNA sequencing detected the Q61H mutation in 52% of colorectal cancer patients who developed resistance to anti-EGFR monoclonal antibody (mAb) treatment (Cetuximab and Panitumumab)[36]. These results

suggest that targeting upstream signaling molecules such as RTKs, SOS, and SHP2 may be ineffective against tumors harboring the KRAS Q61H mutation.

At the atomic level, residue 61, located in switch II, is involved in a direct interaction with SOS. A crystal structure of the RAS:SOS complex (1BKD) shows that SOS Thr935 forms a hydrogen bond with the sidechain amide of Gln61[41] (Fig. 2e), which was captured in the MD simulation. The MD simulations showed that the substitution of Glu to His disrupts this interaction by altering the electrostatic and surface complementarity at the key molecular interface between KRAS and SOS (Fig. 2f). A previous study also reported that mutation of Thr935 to Glu decreased the exchange activity of the yeast homolog of SOS^Cat (SCDC25)[56]. Furthermore, a hydrophobic interaction between KRAS Q61H and SOS_cat was distinctively compromised by the mutation (Supplementary Fig. 6c). Likewise, the Q61H mutation has been reported to disrupt the sensitivity of HRAS to the GEF activity of SCDC25, whereas Q61L had little effect[57]. In addition to the specific sidechain interactions described above, SOS makes extensive interactions with switch I, which exhibits evidence of structural perturbations in the HSQC spectrum of the KRAS Q61H mutant. Clearly, the substitution to His at the 61 position has dramatic effects on GEF-mediated exchange in both KRAS and HRAS. We did not observe any substantial differences in the pattern of Src-mediated phosphorylation or SHP2-mediated dephosphorylation of the KRAS Q61H mutant vs. WT. Similar to WT, KRAS Q61H can be phosphorylated at Tyr32 and Tyr64 by Src, and dephosphorylated by SHP2 (Fig. 3). However, we found that Src phosphorylation has less impact on signaling downstream of KRAS Q61H relative to WT or other KRAS mutants, largely because its binding to RAF-RBD is uncompromised by phosphorylation (Fig. 5a, b). In agreement with our findings, KRAS Q61H was shown to bind more tightly to RAF-RBD than p110γ-RBD[58]. As all three RAF RBDs bind RAS in the same structural mode and the contacting residues are highly conserved[59], we expect the modest effect of KRAS Q61H phosphorylation on affinity for RAF-RBD to extend to other RAF isoforms. However, binding to other effector proteins, in particular those that interact with Switch II regions such as PI3Kγ[60], Nore 1[61], and AGO2[62], may be more strongly impacted.

Phosphorylation favors accumulation of the GTP-loaded state due to enhanced intrinsic nucleotide-exchange rate and impaired GTP hydrolysis. We propose here that resistance to SHP2 inhibitors of KRAS Q61H cells stems from these unique features, which maintain MAPK signaling even when KRAS is phosphorylated, together with SOS GEF and RASA1 GAP insensitivity (Fig. 5d). Various clinical trials of SHP2 inhibitor therapies for solid tumors have taken different approaches to using RAS

mutation status in their eligibility criteria, from no exclusions (e.g., JAB-3068) to excluding specific mutation types (RASQ61X for BBP-398; RASQ61X and KRASG13X for RMC-4630) or all known activating RAS mutations (except KRAS G12X for TNO155). A better understanding of the mechanistic details underlying resistance or sensitivity of each specific Q61X mutation should help guide selection of therapeutic opportunities.

The decoupling of the Q61H mutant from upstream regulation is consistent with the lack of observed efficacy of inhibitors of upstream processes. The present findings suggest that the targeted inhibition of downstream effectors (e.g., RAF, MEK, ERK, PI3K, AKT, and mTOR) or combinations thereof may be a more rational and effective approach for the treatment of cancers driven by KRAS Q61H. As a proof of principle, we treated PDAC cells harboring KRAS G12V or -Q61H mutations with the MEK inhibitor trametinib, the ERK inhibitor ulixertinib, or a combination of both drugs (Supplementary Fig. 18a). Treatment with either single agent resulted in slight inhibition of growth, whereas the combination treatment significantly reduced the viability of PDAC cells with either KRAS mutation, compared to vehicle control. Similar results were observed in HEK293T cells engineered to overexpress KRAS WT, -G12V, or -Q61H (Supplementary Fig. 18b). Concordantly, it was recently found that in lung cancer cells, KRAS Q61H signals more strongly through MAPK than PI3K/AKT, and that these cells responded to RAF and MEK inhibitors[58]. We treated patient-derived organoids with KRAS-G12D or -Q61H mutations with trametinib and the AKT inhibitor MK-2206, and found organoids with either KRAS mutation responded to both inhibitors (Supplementary Fig. 18c). Together, our findings suggest that a rational approach for treating cancers with KRAS Q61H mutation should consider the inhibition of targets downstream of KRAS.

The GTPase cycle of RAS proteins is often described as a two-state ON/OFF "switch" by virtue of definitive dependency on the nucleotide bound (GTP/GDP) and the remarkable acceleration of nucleotide cycling mediated by GEF and GAP activities. In our model of the modulation of KRAS signaling, the regulator-dependent rapid ON/OFF switching function is largely compromised by tyrosyl phosphorylation and the system becomes highly dependent on intrinsic nucleotide exchange and hydrolysis activities, as well as SHP2-mediated tyrosyl dephosphorylation, which are slower processes. Due to the impairment of effector binding caused by phosphorylation, the Src/SHP2 regulation of KRAS signaling can be seen as a "dimmer" in series with the nucleotide-dependent ON/OFF switch: when KRAS is switched "ON" by bound GTP, tyrosyl phosphorylation by Src can "dim" its signaling output, whereas dephosphorylation by SHP2 restores signaling. The Q61H mutation effectively short circuits this system by locking the switch "ON" and bypassing the dimmer, as this mutant does not evade phosphorylation by Src, but the posttranslational modification has little impact on its ability to signal. In contrast, KRAS G12 mutations cause the switch to stick in the ON position, but signaling from these mutants can still be downregulated by the dimmer (i.e., phosphorylation); thus, they are sensitive to SHP2 inhibitors. Thus, RAS signaling output is finely tuned by both a nucleotide-dependent ON/OFF binary switch and a tyrosyl phosphorylation-dependent "dimmer" switch, and both switch functions appear to be impacted by specific mutations.

## Methods

**Cells**. HEK293, CFPAC1, Capan-1, HPAF-II, SW1990, HUPT3, MiaPaCa-2, and Hs766T cells were obtained from the American Type Culture Collection. P411T1 were generated from PDAC PDXs[63]. OCIP.236, OCIP.347, and PPTO.93 were generated by the Princess Margaret Living Biobank Organoid core facility

(https://www.livingbiobank.ca, Toronto, Canada). HEK293, MiaPaCa-2, and Hs766T cells were maintained in Dulbecco's modified Eagle's medium (DMEM; Invitrogen) supplemented with 10% (v/v) heat-inactivated fetal bovine serum (FBS; Wisent) at 37 °C in a humidified 5% $CO_2$ atmosphere. CFPAC1, Capan-1, HPAF-II, SW1990, HPAC, and HUPT3 cells were maintained similarly in RPMI-1640 (Wisent) medium supplemented with 10% (v/v) FBS. P411T1 cells were maintained in DMEM/F12 medium (Thermo Fisher Scientific, 11330-032) supplemented with 5 ng/ml EGF (R&D Systems, 236-EG-01M), 10 μg/ml insulin (Thermo Fisher Scientific, 12585-014), and 10% (v/v) FBS.

**Plasmids for mammalian cell expression**. A plasmid cDNA encoding human KRAS4B WT was subcloned from a pBabe plasmid (generously provided by Dr. Channing Der, University of North Carolina, Chapel Hill) into pcDNA3. KpnI and NotI restriction enzymes were used to integrate an N-terminal HA tag to the plasmid. We conducted site-directed mutagenesis (Quikchange; Roche) to generate single and double KRAS mutants (G12V, G12D, Q61H, Y32F/Q61H, and Q61H/Y64F). pCMV5-Src (WT or K295RY527F), pCMV5-SHP2 (WT and C459S), pCGN-HA-HRAS WT, pCGN-HA-KRAS WT, and pCGN-HA-NRAS WT were obtained from Addgene. Gateway Cloning technology (Invitrogen) was used to subclone Flag-SHP2 constructs into pcDNA3. All the plasmids were verified by direct DNA sequencing before they were used in this study.

**Antibodies**. A 12-mer phospho-peptide comprising the KRAS sequence flanking pTyr64 (TAGQEEpYSAMRD) was used to generate anti-pRAS Y64 rabbit polyclonal antibody (1 : 1000; Bethyl Laboratory, Inc.). Pan-Ras (OP40, 1 : 500), mAbs against HA (12CA5, 1 : 500), KRAS (OP24, 1 : 200, 1 : 500), and pTyr (4G10) (05-321, 1 : 1000) were obtained from Boehringer Ingelheim and Millipore, respectively. Monoclonal FLAG-M2 (F1804, 1 : 2000), β-actin (A5316, 1 : 2000) and Vinculin (V9264, 1 : 2000) antibodies were obtained from Sigma. Rabbit polyclonal antibodies against Src (#2109, 1 : 5000), pERK (#9101, 1 : 1000), ERK (#9102, 1 : 1000), pTyr (P-Tyr-100) (#8954, 1 : 2000), Ras (3965 S, 1 : 1000), and HA (#3724, 1 : 5000) were obtained from Cell Signaling Technologies. HA.11 (#16B12, 1 : 1000) was obtained from Covance. Polyclonal IgG (sc-2027), HRAS (sc-520, 1 : 1000), NRAS (sc-519, 1 : 1000), and SHP2 (sc-280, 1 : 1000) were obtained from Santa Cruz Biotechnology.

**Chemicals**. SHP099, Trametinib, and Ulixertinib were obtained from Selleck Chemicals. MK-2206 was obtained from Cayman Chemicals. EGF was obtained from R&D Systems. Compound 11a-1, 6-Hydroxy-3-iodo-1-methyl-2-(3-(2-oxo-2-((4-(thiophen-3-yl) phenyl)amino)acetamido)phenyl)-1H-indole-5-carboxylic acid was synthesized as described[37].

**CRISPR/Cas9-mediated gene editing**. Gene editing was conducted using pLentiCRISPR (49535) from Addgene, with a guide sequence derived from exon 1 of SHP2 (human) and the non-target control sequence (Supplementary Table 5)

**Lentivirus production and infection of cell lines**. Lentivirus was produced by transfecting HEK293FT cells (Thermo Fisher Scientific) with psPAX2, pMDG1.vsvg, and pLentiCRISPR transfer vector. The supernatant, which contains lentivirus, was collected at 72 h post transfection and filtered before being used[64]. Efficient retroviral infection required addition of 5 μg/ml Polybrene (Millipore) for HEK293, CFPAC1, HPAF-II, and Hs766T. Selection was performed using puromycin (5 μg/ml) (Wisent) or hygromycin (100 μg/ml) (Invitrogen) 24 h after infection and polyclonal populations were generated.

**Immunoprecipitation and immunoblotting**. Immunoprecipitation and western blotting were performed as follows[65]. Cells were collected in EBC lysis buffer (50 mM Tris pH 8, 120 mM NaCl, 0.5% NP-40) supplemented with protease and phosphatase inhibitors (Roche). Lysates were immunoprecipitated using the indicated antibodies along with protein A-Sepharose (Repligen). Bound proteins were washed five times in NETN buffer (20 mM Tris pH 8, 100 mM NaCl, 1 mM EDTA, 0.5% NP-40), eluted by boiling in sample buffer and resolved by SDS-polyacrylamide gel electrophoresis (PAGE). Proteins were electrotransferred onto polyvinylidene difluoride membrane (Bio-Rad), blocked, and probed with the indicated antibodies.

**Cellular RAS/RAF-RBD-binding assay**. RAS/RAF binding was assessed using the RAS activation assay kit from Millipore (17-218). Briefly, KRAS-GTP protein was pulled down from lysates of cells treated under various conditions using an agarose-bound Glutathione S-transferase (GST) fusion protein corresponding to the RBD of human RAF1. KRAS bound to RAF-RBD was detected by western blotting using an anti-RAS antibody (Millipore #05-516, 1 : 2000).

**Cell proliferation assay**. Equal numbers of cells were plated in quadruplicate in 96-well plates in the presence of a range of concentrations of the indicated inhibitors for 48 h and cellular proliferation was assessed using Alamar Blue proliferation assay as per the manufacturer's instructions (Invitrogen #DAL1100).

**3D spheroid assay**. In vitro 3D spheroid culture was performed using an Ultra-Low Attachment surface-coating 96-well spheroid microplate (Corning). Briefly, equal numbers of cells were plated in 96-well spheroid microplates and cultured in the presence or absence of indicated inhibitors. Spheroid viability was determined using the CellTiter-Glo 3D Cell viability assay according to the protocol provided by the manufacturer.

**Organoid culture and drug screening**. Organoids were propagated in hPDAC media modified from Boj et al.[66]. For drug screening, organoids were dissociated to single cells and seeded on top of a thin layer of Matrigel (Corning, NY, USA) in 384-well plate (3000/well). Next day, organoids were treated with a range of MK-2206 (0.05–30 μM) and Trametinib (0.001 nM to 10 μM) or SHP099 (0.05–400 μM) concentrations for 96 h and cell viability was determined by CellTiter-Glo 3D viability assay (Promega, Madison, USA). Drug-response curves were graphed and half maximal inhibitory concentration values were calculated using GraphPad Prism 8.0 (San Diego, USA).

**Recombinant protein expression**. A synthetic gene encoding KRAS4B residues 1–173 (with C118S mutation) optimized for *Escherichia coli* expression (GenScript) was cloned into a pET-28 vector and expressed in *E. coli* (BL21) (residue 1–173, with C118S mutation). QuikChange site-directed mutagenesis was used to produce specific KRAS mutants (G12V, G12C, G12D, G13D, Q61H, and Q61L, Y32F, Y64F, Y40F) (Supplementary Table 5). The $^{15}$N-labeled proteins were expressed by culturing the bacteria in M9 minimal media supplemented with 1 g/L of 15 ammonium chloride and at 37 °C until the OD at 600 nm reached 0.6. Protein expression was induced with 0.2 mM IPTG (isopropyl β-D-1-thiogalactopyranoside) at 16 °C overnight. The culture was centrifuged and the cell pellets were re-suspended with lysis buffer to collect (50 mM Tris, 150 mM NaCl, 0.1% NP-40, 10% Glycerol, 10 mM Imidazole, 5 mM MgCl$_2$, 1 mM phenylmethylsulfonyl fluoride, 10 mM β-mercaptoethanol, and lysozyme at pH 8.0) and lysed by sonication. Cell debris was removed by centrifugation (23,708 × g for 45 min) and His-tagged KRAS was purified by Ni-NTA resin. The His tag was cleaved by thrombin and the protein was purified using size-exclusion chromatography (SEC; Superdex$^{TM}$ S75 (GE Healthcare) 26-60 column with 20 mM HEPES, 100 mM NaCl, 5 mM MgCl$_2$, and 2 mM tris(2-carboxyethyl)phosphine (TCEP) pH 7.4).

The kinase domain of Src was produced as a recombinant His-tagged protein in *E. coli*[67]. Briefly, cDNA encoding to Src residues 254–536 was subcloned into pET-46 Ek/LIC. In addition, DNA corresponding to full-length *Yersinia pestis* YopH phosphatase was subcloned into pRSF Ek/LIC. In both constructs, a thrombin protease site was introduced after the vector encoded N-terminal 6-histidine purification tag. Expression of soluble Src in the Rosetta-2 (DE3) *E. coli* strain was achieved by co-expression of YopH to mitigate toxicity of Src kinase activity. Cell cultures were grown to an OD$_{600}$ = 0.8 at 37 °C and recombinant protein expression was induced with a final concentration of 0.5 mM IPTG for 20 h at 18 °C. Pelleted cells were re-suspended in 50 mM Tris-HCl pH 8.0, 500 mM NaCl, 25 mM imidazole, and 5% (v/v) glycerol, lysed using a hydraulic cell disruption system (Constant Systems), and recombinant protein was purified by standard Ni-NTA affinity chromatography. Pooled elutions containing both Src and YopH were then dialyzed overnight against 20 volumes of buffer containing 20 mM Tris-HCl pH 8.0, 100 mM NaCl, 5% (v/v) glycerol, and 1 mM dithiothreitol (DTT). Following overnight dialysis, the protein solution was applied onto an anion exchange column (Mono Q 5/50 GL) equilibrated with 20 mM Tris-HCl pH 8.0, 5% (v/v) glycerol, and 1 mM DTT (Buffer A). Protein bound to the anion exchange column was eluted with a 100 CV gradient of 0–50% Buffer B (20 mM Tris-HCl pH 8.0, 1 M NaCl, 5% glycerol, and 1 mM DTT). Anion exchange was sufficient for the separation of recombinant Src kinase from the YopH phosphatase. The elutions containing Src kinase were pooled and further purified by SEC using a custom Superdex-200 10/300 prep grade column equilibrated with 50 mM Tris-HCl pH 8.0, 100 mM NaCl, 5% (v/v) glycerol, and 1 mM DTT. All purification steps were carried out at 4 °C. Protein concentration was determined by absorbance at $λ$ = 280 nm and purity was confirmed by SDS-PAGE and MS.

cDNA sequences encoding human SHP2 were cloned into pET-46 Ek/LIC. A thrombin protease site was introduced downstream of the 6-histidine purification tag. Constructs were verified by DNA sequencing. SHP2 was expressed in *E. coli* BL21 (DE3) cells. Cell cultures were grown to an OD$_{600}$ = 0.8 at 37 °C and recombinant protein expression was induced with a final concentration of 0.5 mM IPTG for 20 h at 18 °C. Cells were lysed in 50 mM Tris-HCl pH 8.0, 500 mM NaCl, 25 mM imidazole, and 5% (v/v) glycerol. Proteins were purified by Ni-NTA affinity chromatography. Pooled elutions were dialyzed overnight against buffer containing 20 mM Tris-HCl pH 8.0, 100 mM NaCl, 5% (v/v) glycerol, and 2 mM DTT. Following overnight dialysis, the protein solution was further purified on an anion exchange column (Mono Q 5/50 GL) equilibrated with 20 mM Tris-HCl pH 8.0, 5% (v/v) glycerol, and 2 mM DTT. Fractions containing the protein were pooled and loaded on a Superdex-200 10/300 column equilibrated in 50 mM Tris-HCl pH 8.0, 100 mM NaCl, 5% (v/v) glycerol, and 2 mM DTT. Purified protein was stored at 2 mg/ml.

The (RBD) of BRAF (cDNA encoding residues 150–233) was subcloned into pGEX-4T2 (GE Healthcare) to produce an RBD bearing an N-terminal GST tag. The cDNA encoding GAP domain of human RASA1 (residues 715–1074) and the catalytic domain of SOS$^{cat}$ (residues 564–1049) were subcloned in to pET15b

(Novagen/EMD Biosciences). These proteins were expressed in *E. coli* BL21 (DE3). The His tag was cleaved by thrombin followed by final purification by SEC (Superdex$^{TM}$ S75, GE Healthcare).

**In vitro kinase assay**. Purified recombinant human KRAS was incubated with purified recombinant active His-tagged Src kinase domain in kinase buffer (50 mM HEPES pH 7.5, 10 mM MgCl$_2$, 1 mM EGTA, 0.01% Brij-35, 200 μM ATP) for 1 h at room temperature. Thereafter, proteins were denatured by boiling in sample buffer and resolved by SDS-PAGE. Phosphorylation was detected by anti-pTyr and anti-RASpTyr64.

**Real-time NMR GTPase assay**. Before performing real-time NMR-based nucleotide-exchange assays, the WT and mutant KRAS samples were confirmed to be GDP loaded on the basis of their $^1$H-$^{15}$N HSQC spectrum. Then, 40 μl of 250 μM GDP-loaded $^{15}$N-KRAS sample was incubated with 10× molar excess (2.5 mM) GTPγS (guanosine 5′-[γ-thio] triphosphate, tetralithium salt, Sigma-Aldrich) in a 1.7 mm NMR tube. Sequential $^1$H-$^{15}$N HSQC spectra were collected for the time course of the exchange reaction using a Bruker 600 MHz Avance III NMR spectrometer equipped with a 1.7 mm cryogenic TCI MicroCryoProbe. For the GEF assays, catalytic domain of SOS (residues 564–1049 SOS$^{cat}$) was added at a molar ratio of 1 : 600 to KRAS. To perform GTP hydrolysis assays, $^{15}$N-KRAS samples were loaded with GTP by incubation with a tenfold excess of GTP in the presence of EDTA. The sample was then passed through gel filtration chromatography (Superdex 75 10/300, GE Healthcare) to remove excess nucleotide and sequential HSQC spectra were collected as hydrolysis proceeds. The GAP assays were performed by adding recombinant GAP domain of RASA1 (GAP-334) at a 1 : 3000 molar ratio to KRAS. The tyrosyl-phosphorylated $^{15}$N-KRAS Q61H sample was prepared by incubating the GDP-loaded mutant with a catalytic amount of recombinant Src (i.e., 1 : 250 molar ratio) in the presence of 2 mM ATP, 1 mM activated sodium vanadate, 2 mM imidazole, 1 mM sodium fluoride, and 1.15 mM sodium molybdate. Phosphorylation was confirmed using $^1$H-$^{15}$N HSQC spectrum and before the excess ATP was removed from the KRAS sample by gel filtration chromatography (Superdex$^{TM}$ S75 GE Healthcare).

NMR data were processed using NMRPipe and analyzed using NMRFAM-SPARKY software[68,69]. The nucleotide-exchange and hydrolysis rates were calculated based on the fraction of GDP-bound KRAS peak intensities obtained from at least three cross-peaks, which were plotted as a function of time and fit to a one-phase exponential association curve[40] using GraphPad Prism 7.

**Mass analysis of intact pKRAS**. To obtain accurate masses of KRAS Q61H and pKRAS Q61H samples in 20 mM HEPES, 100 mM NaCl, 5 mM MgCl$_2$, 2 mM TCEP, 1 mM activated sodium vanadate, 2 mM imidazole, 1 mM sodium fluoride, and 1.15 mM sodium molybdate at pH 7.4 were diluted 1 : 5 with 20 mM Tris-Base, 5 mM MgCl$_2$, 2 mM TCEP pH 5.5, to obtain a final protein concentration of 50 μM. Mass spectra were obtained on an Agilent 6538 Ultra High Definition Quadrupole time-of-flight mass spectrometer. The experiment was run in positive mode with electrospray ionization.

**Ion-exchange chromatography**. The mono-, di-, and unphosphorylated forms of KRAS WT and Q61H were separated by anion exchange chromatography using a Mono Q 5/50 GL column run with 20 mM HEPES pH 7.0, 5 mM MgCl$_2$, and 1 mM TCEP (Buffer A), and 20 mM HEPES pH 7.0, 5 mM MgCl$_2$, 1 mM TCEP, and 1 M NaCl (Buffer B). The separation of the three forms was achieved with an 80 column-volume gradient of 0–40% buffer B. To maintain the phosphorylation of the samples, phosphatase inhibitors were added to the wash and elution buffer (1 mM activated sodium vanadate, 2 mM imidazole, 1 mM sodium fluoride, and 1.15 mM sodium molybdate).

**Biolayer interferometry**. An Octet RED-384 biolayer interferometry instrument equipped with the Octet Data Acquisition 9.0.0.37 and FortéBio Data analysis software (Pall) was used to measure the affinity of mono-, di-, and unphosphorylated samples of KRAS (WT and Q61H). GST-tagged BRAF-RBD (residues 150–233) was immobilized on anti-GST-conjugated biosensors (Pall FortéBio) by incubating the sensor with 5 μg/ml of BRAF-RBD for 10 min. The sensor was then dipped into wells containing increasing concentration of the KRAS for 30 s. The sensor was then dipped in 20 mM HEPES, 100 mM NaCl, 5 mM MgCl$_2$, and 2 mM TCEP supplemented with 0.5% bovine serum albumin (BSA) and 0.05% buffer to monitor dissociation of KRAS from BRAF-RBD.

The assay was performed using 96-well plates at 25 °C with 1000 r.p.m. agitation in a buffer comprising 20 mM HEPES, 100 mM NaCl, 5 mM MgCl$_2$, and 2 mM TCEP supplemented with 0.5% BSA and 0.05% Tween-20 to minimize nonspecific binding, as well as phosphatase inhibitors (1 mM activated sodium vanadate, 2 mM imidazole, 1 mM sodium fluoride, and 1.15 mM sodium molybdate). BRAF-RBD immobilized sensor dipped into buffer alone was used as a control. Binding data were fit to a one-to-one binding stoichiometry model using steady-state analysis.

**Liquid chromatography and mass spectrometry**. Following phosphorylation by Src kinase domain, KRAS Q61H samples were reduced and alkylated with DTT (5 mM) and iodoacetamide (10 mM), and digested with trypsin (10 μg/ml in 50 mM NH$_4$HCO$_3$ pH 8.3) at 37 °C for 16 h. The resultant tryptic peptides were de-salted using reverse-phase C18 columns and lyophilized in a vacuum centrifuge. The dried samples were reconstituted in 0.1% HCOOH and analyzed by LC-MS. An EASY-nLC 1200 pump was used to load and resolve samples on an Acclaim PepMapTM 100 nanoViper pre-column (75 μm × 2 cm, 3 μm) and an in-line Acclaim PepMapTM RSLC nanoViper analytical column (75 μm × 50 cm, 3 μm) with a gradient of 5–30% acetonitrile over 120 min in a 0.1% HCOOH mobile phase. Positive-mode electrospray ionization was applied and ions were analyzed by MS using a Q-Exactive HF instrument set to perform MS/MS HCD fragmentation scans on ≤20 most intense ions (ion count ≥ 000 for activation) from an MS parent ion scan (390–1800 m/z range; 60,000 full-width half-maximum resolution @200 m/z). Fragmented ions were placed on a dynamic exclusion list for 5 s. Acquired raw files were converted to mzML format using Proteowizard (v3.0.10800), then searched using X!Tandem (v2013.06.15.1) against Human RefSeq Version 45 (36,113 entries). The search parameters specified tolerances of 15 p.p.m. for the parent ion and 0.4 Da for fragment ions. One missed trypsin cleavage was allowed and carbamidomethylation [C] was set as a fixed modification, whereas oxidation [M], deamidation (N, Q), acetylation (protein N-term), and phosphorylation (STY) were allowed as variable modifications. All data are publicly available through the MassIVE archive (https://massive.ucsd.edu).

**MD simulations**. MD simulations were conducted using NAMD 2.13[70]. The revised CHARMM36 force field[71–73] was used for the proteins and CHARM-GUI server[74] was used to parameterize the nucleotide. The starting coordinates for the MD simulations were obtained from crystal structures deposited in the PDB under the following accession codes: 6GOG and 6GOD[75], 1WQ1[76], 4G0N[8], 4OBE[77], and 6EPL[78]. The details of each system are presented in Supplementary Table 3. The MD simulations were conducted following previously published approaches with some modification[51,79]. At the preparation phase, each system was placed in a solvent-filled cube of TIP3P water molecules with a solvent padding size of 20 Å in each dimension. NaCl (150 mM) was then added to each system to maintain physiological ion concentrations and neutralize the system. To prepare each system for simulation, first the energy of the solvent and protein systems were minimized using 5000 steps of conjugate gradient energy minimization[80]. The systems were then heated to 310 K at a rate of 0.2 K/fs, whereas a harmonic restraint force constant of 4.0 kcal mol$^{-1}$ Å$^2$ was applied to the α-carbons of the protein and the heavy atoms of GTP. Then, equilibration was achieved by slowly reducing the restraint force constant to zero at a rate of 0.01 kcal mol$^{-1}$ Å$^2$ ps$^{-1}$. Finally, the systems were further equilibrated for 10 ns under a constant temperature of 310 K and pressure of 1 atm[17]. During both the preparation and production simulations, we used a non-bonded pair list cutoff of 14 Å and 20 fs update time with particle mesh Ewald algorithm[81]. Constant pressure (1 atm) was maintained using the Nose–Hoover Langevin piston method[82] with a period of 100 fs and a decay interval of 50 fs, whereas Langevin dynamics algorithm was applied to maintain constant temperature (310 K). The short-range non-bonded interactions were switched off between 10 and 12 Å. Periodic boundary conditions were applied in all simulations. Each system was simulated for three sequential 300 ns periods, with three different starting velocities, to generate a total simulation time 900 ns. VMD 1.9.4[83] and Bio3d[84] were used for data analysis and model generation.

**Statistical analyses**. All experiments were performed in triplicate with mean and SE reported. Unpaired two-tailed Student's t-test was used to compare between treatment groups and cell types. All statistical analysis was performed using GraphPad PRISM 4.0 or 7.0 software. P-value < 0.05 was considered statistically significant.

**Reporting summary**. Further information on research design is available in the Nature Research Reporting Summary linked to this article.

## Data availability
Mass spectrometry data are available in the MassIVE repository under accession number MSV000084657. All other data supporting this study are included in the article and the supplementary files. Source data are provided with this paper.

## Code availability
The study utilized all-atom molecular dynamics simulation codes as described in CHARM-GUI[74] https://www.charmm-gui.org/. The details of the simulation parameters are described in the "Methods" section.

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

## Acknowledgements

This work was supported by funds from the Canadian Cancer Society Research Institute (703209 to M.I.), Princess Margaret Foundation (M.I.), Canadian Institutes of Health Research (MOP-133694 to JEL, FDN-1542284 to M.I. and PJT-166005 to M.O.), NIH RO1 CA207288 (Z.-Y.Z.). M.I. holds a Canada Research Chair in cancer structural biology. T.G. was supported by the George and Helen Vari Foundation (Toronto, Canada) and Y.K. was supported by the Program for Advancing Strategic International Networks to Accelerate the Circulation of Talented Researchers from Japan Society for the Promotion of Science and a Long-Term Overseas Research Fellowship from

Sumitomo Life Welfare and Culture Foundation. NMR spectrometers were funded by the Canada Foundation for Innovation (CFI) and the NMR Core Facility is supported by the Princess Margaret Cancer Foundation. MD simulations were enabled in part by resources provided by Compute Ontario and Compute Canada (www.computecanada.ca). We thank Dr. Geneviève Seabrook for technical expertise and access to the NMR Core Facility.

## Author contributions

T.G., Y.K., C.B.M., M.O., and M.I. conceptualized the project and wrote the manuscript. T.G., Y.K., J.S.-G., N.R., M.L.U., R.H., B.P.K.P., W.H., I.V.-S., C.M.R., and M.H. conducted experiments. A.M. provided technical expertise for MD simulations. J.M., Z.-Y.Z., and J.J.Y. provided reagents or cell lines. C.B.M., M.O., M.I., M.S.I., J.E.L., M.-S.T., and B.R. supervised. All authors reviewed and edited the manuscript.

## Competing interests

The authors declare no competing interests.
