## [Peer Review File · Nature Communications]

The Q61H mutation decouples KRAS from upstream regulation and renders cancer cells resistant to SHP2 inhibitorsReviewers' comments:

Reviewer #1 (Remarks to the Author):

The authors present a report detailing the mechanisms by which KRAS Q61H confers resistance to SHP2 inhibitors. The major claim is that resistance occurs because Q61H does not rely upon SOS1 for nucleotide exchange and that the combination of impaired GAP-stimulated and intrinsic GTP hydrolysis and phosphorylation-dependent enhanced nucleotide exchange leads to accumulation of GTP-bound protein. They also introduce a new antibody against pY64 that will be useful to other researchers. While the rarity of the Q61H mutation may limit the impact on translation, this work is still valuable because it provides fundamental mechanistic information as to how RAS works. The paper is well written and the figures are generally well presented. I support publication of this work if comments below can be answered.

Specific comments:

"cell lines harboring the KRAS Q61H mutation are markedly less sensitive" overstates the difference in GI50. There appears to be a statistical difference but the fold change is modest.

S1: Dose response curves are typically presented with log scale in the x axis. Would do this way.

It is also confusing which panels belong to which panel letters. Would make that more clear.

Fig 1e. Inhibition of pERK for KRAS WT is not very convincing

Fig 1f. Is it established that this line of HEK293 cells overexpressing KRAS are dependent on KRAS for viability?

Fig 4: It seems highly plausible that phosphorylation of Y64 will increase the exchange rate of most RAS proteins. I would be surprised if this is particular to Q61H. Has that already been shown? Fig 5d implies that it has. Some comment on this in the manuscript would be helpful.

Fig 5b: To me, the amount of KRAS Q61H pulled down does appear to decrease somewhat with increasing SRC. However difficult to interpret with provided exposure.

Reviewer #2 (Remarks to the Author):

In this study, the authors explore the basis for SHP2 inhibitor resistance that is shown upon expression of KRAS Q61H, in contrast to other mutant allelotypes of KRAS such as G12C or G12V. The authors develop a model based on a balance of SHP2 and SRC mediated regulation that shows unique biochemical features in the context of KRAS Q61H in which the GTP loaded state of the oncoprotein is maintained and uncoupled from regulatory proteins.

The data are interesting and provide insight. However, there are several issues that diminish enthusiasm and the findings remain preliminary.

(1) Most experiments are performed with over-expressed proteins of interest. The key findings need to be validated using cells expressing endogenous levels.

(2) Key mechanistic details are lacking. For instance, it is speculated that GTP-loading of RAS mutants engaging effector proteins would be protected from Src kinase activity. However, this is not experimentally validated.

(3) Does Src inhibition enhance KRAS Q61H GEF/GAP sensitivity and SHP2 inhibitor sensitivity?

(4) In figure 1, RAS-GTP levels need to be shown for panels c-e.

(5) In figure 2, how do the authors explain why G13D shows higher intrinsic rate of exchange than WT (panels b, d), and why Q61L is distinct from Q61H (panels b, d)? In panel e, KRAS Q61L should be included for control.

(6) In figure 3, panel g should contain also a catalytic impaired mutant of SHP2 for control.

(7) In figure 5, the model is not clear and contains contradictions. For instance, the authors do not

take into account the stoichiometry of SRC and SHP2 activities on WT versus mutant KRAS G61H. Furthermore, is there a quantitative difference in pERK output depending upon the balance between SHP2 and SRC activities? Finally, RAF is referred to generically yet the authors test BRAF RBD more specifically. Why do the authors focus on BRAF RBD and not the more commonly used probe of CRAF RBD and do the findings generalize to all RAF isoforms?

(8) The therapeutic impact of the model is not experimentally validated. What is the most appropriate approach to restoring sensitivity of KRAS G61H to upstream pathway blockade (via SHP2 inhibition) and inhibit cancer cell signaling and viability? This needs to be experimentally shown.

Reviewer #3 (Remarks to the Author):

The team of Marshall, Ohh and Ikura present a study of the oncogenic Q61H mutant of K-Ras4B. The work provide evidence that SHP2 inhibition have no substantial effect on inhibiting the function of this Ras mutant downstream and furthermore the authors demonstrate this Ras mutant no longer interacts with a key exchange factor protein, SOS1 or GAP protein p120.

Overall this study is well executed and within the ground covered by the authors quite complete. However, I would strongly encourage to expand their study to the molecular / structural level by carrying out now relatively standard model-building for Ras Q61H : SOS1, : p120 and Raf-RBD complexes and perhaps a reasonable all-atom molecular dynamics (~ 300 ns) carried in quadruplicate to equilibrate these models. (the crystal structure of the Q61H is recently become available; one in 2017, one in 2019). In other words what is shown as Fig. 5c should be significantly expanded to the interactions with p120 and Raf-RBD Models with the RBD should probably be run with and without Ras phosphorylation. Only if such data re presented, I would find the study compelling and complete at the molecular level.

Minor concerns:

- 1) The authors should mention more prominently that Sos1 and p120 are just two of several regulatory proteins that may act on Q61H in cancer cells. Similarly concerning Raf as one of the effector proteins.
- 2) The work of Cruz-Migoni, A et al., PNAS 2019 and Gupta, AK et al., Chem.Biol Drug Des 2019 should be cited.
- 3) An overview figure how SHP2 relates to Sos1 and p120 as well as MAPK signaling should be given at the beginning of the paper.
- 4) While the real time NMR assays are a critical strength of the paper, it would be good to remind the reader why the curves do not go to zero (e.g. Fig. 2a,c) and different mutants have different off-sets. Also why GTPgammaS is used, rather than GMPPNP in these assays.

RESPONSE TO REVIEWER #1:

The authors present a report detailing the mechanisms by which KRAS Q61H confers resistance to SHP2 inhibitors. The major claim is that resistance occurs because Q61H does not rely upon SOS1 for nucleotide exchange and that the combination of impaired GAP-stimulated and intrinsic GTP hydrolysis and phosphorylation-dependent enhanced nucleotide exchange leads to accumulation of GTP-bound protein. They also introduce a new antibody against pY64 that will be useful to other researchers. While the rarity of the Q61H mutation may limit the impact on translation, this work is still valuable because it provides fundamental mechanistic information as to how RAS works. The paper is well written and the figures are generally well presented. I support publication of this work if comments below can be answered.

Specific comments:

1. “cell lines harboring the KRAS Q61H mutation are markedly less sensitive” overstates the difference in GI50. There appears to be a statistical difference but the fold change is modest.

Response:

We have removed the word ‘markedly’.

2. S1: Dose response curves are typically presented with log scale in the x axis. Would do this way. It is also confusing which panels belong to which panel letters. Would make that more clear.

Response:

We thank the Reviewer for this suggestion. We have accordingly changed the x-axes to log scale (Suppl Fig. S1).

3. Fig 1e. Inhibition of pERK for KRAS WT is not very convincing

Response:

Note: this is now Fig. 1f. SHP2 inhibitor treatment markedly attenuated pERK levels in cells ectopically or endogenously (i.e., mock control) expressing KRAS WT, but had negligible effect on pERK levels in cells ectopically expressing KRAS Q61H. To better illustrate these differences, we have measured the pERK signal intensities in a semi-quantitative manner using densitometry normalized to the vinculin loading control (Revised Fig. 1f).

4. Fig 1f. Is it established that this line of HEK293 cells overexpressing KRAS are dependent on KRAS for viability?

Response:

Note: this is now Fig. 1g and Suppl Fig S3). The purpose of this experiment was to ask whether cells could be rendered less sensitive to SHP2 inhibition by ectopically expressing KRAS Q61H in comparison to control isogenic cells expressing KRAS WT. Whether this particular cell line is dependent on KRAS for survival is not formally known; however, KRAS knockout, unlike H- or N-RAS, in mice is embryonic lethal (PMID: 9334313), suggesting its essentiality in development. Additionally, KRAS-transfected HEK293 cells have been widely used in the RAS community in a range of experiments including characterizing the impact of allele-specific KRAS G12C inhibitors (PMID: 26841430), and interactions with binding partners (PMID: 26854235 and 26959608), supporting the validity of this tool.

5. Fig 4: It seems highly plausible that phosphorylation of Y64 will increase the exchange rate of most RAS proteins. I would be surprised if this is particular to Q61H. Has that already been shown? Fig 5d implies that it has. Some comment on this in the manuscript would be helpful.

Response:

Yes, we have previously shown (PMID: 30644389) that phosphorylation increases the exchange rate of WT and G12V KRAS proteins. However, whereas WT and G12V are also sensitive to SOS-assisted nucleotide exchange, the Q61H mutant is distinct in that it is resistant to the GEF activity of SOS, and only phosphorylation enhances its nucleotide exchange. We have added comments to clarify this point in the discussion section (page 11).

6. Fig 5b: To me, the amount of KRAS Q61H pulled down does appear to decrease somewhat with increasing SRC. However difficult to interpret with provided exposure.

Response:

This is an astute observation by the Reviewer, and we agree that there is some reduction. However, the decrease in KRAS Q61H with RBD-pull down (~15%) is modest in comparison to KRAS WT which exhibited a more dramatic reduction (~70%) with increasing concentration of SRC. To better illustrate this observation, we have used densitometry to provide semi-quantitative measurements (Revised Fig. 5b). This is consistent with the statement and model that describes the impact of phosphorylation on RBD binding is less pronounced for Q61H than WT and G12V.

RESPONSE TO REVIEWER #2:

In this study, the authors explore the basis for SHP2 inhibitor resistance that is shown upon expression of KRAS Q61H, in contrast to other mutant allelotypes of KRAS such as G12C or G12V. The authors develop a model based on a balance of SHP2 and SRC mediated regulation that shows unique biochemical features in the context of KRAS Q61H in which the GTP loaded state of the oncoprotein is maintained and uncoupled from regulatory proteins. The data are interesting and provide insight. However, there are several issues that diminish enthusiasm and the findings remain preliminary.

1. Most experiments are performed with over-expressed proteins of interest. The key findings need to be validated using cells expressing endogenous levels.

Response:

We thank the reviewer for this comment. Firstly, we have previously shown that endogenous KRAS is tyrosyl-phosphorylated in mouse embryonic fibroblasts, human pancreatic cancer cell lines and pancreatic tumour xenografts using Mn²⁺ PhosTag SDS-PAGE (Kano et al., 2019 Nat Commun). Following this earlier study, we further examined the phosphorylation status of KRAS in more detail using overexpression systems, which enabled us to explore many important questions. As such, we showed here that cancer cells harbouring endogenous KRAS Q61H is more resistant to SHP2 inhibitor treatment. We also showed that the endogenous KRAS Q61H is also tyrosyl-phosphorylated in a human pancreatic cancer cell line, in the presence of EGF treatment (New Figure 3i). Moreover, in isogenic pancreatic cancer cell lines with CRISPR-Cas9-mediated knockout of SHP2, tyrosyl-phosphorylation of endogenous KRAS Q61H was

observed even in the absence of EGF treatment, which is consistent with the notion that SHP2 is involved in de-phosphorylating endogenous KRAS (New Figure. 3i).

2. Key mechanistic details are lacking. For instance, it is speculated that GTP-loading of RAS mutants engaging effector proteins would be protected from Src kinase activity. However, this is not experimentally validated.

Response:

In the revised manuscript we presented new data that show binding of the BRAF RBD to GTP analog-loaded KRAS inhibits Src phosphorylation (Supplementary Figure S12). This is consistent with our prediction that the GTP-dependent interaction of the RBD with the switch regions of KRAS would block access of Src to its substrate residues Y32 and Y64.

3. Does Src inhibition enhance KRAS Q61H GEF/GAP sensitivity and SHP2 inhibitor sensitivity?

Response:

Thanks for the interesting question. The situation is complicated but we think the answer is No. First, our data showed that phosphorylation of KRASQ61H does not affect the GEF/GAP assisted nucleotide exchange/hydrolysis, because this mutant is already resistant to the activities of these regulators (Fig 2d,g). The impact of phosphorylation on RAF RBD binding is also significantly less disruptive compared to the WT. These data suggest that hindering KRAS Q61H phosphorylation by Src inhibition, or alternately, maintaining the phosphorylation of KRAS Q61H by SHP2 inhibition would have minimal effects on the GTPase cycle or the RBD binding of this mutant. This is consistent with our cell-based observations using SHP2 inhibition, whereas we have not tested Src inhibition.

4. In figure 1, RAS-GTP levels need to be shown for panels c-e.

Response:

We respectfully disagree with this suggestion as this is not the purpose of this figure. Figure 1 addressed the question of whether SHP2 inhibitor treatment affected the viability and pERK levels in various cancer cell lines expressing various oncogenic KRAS mutants. This initial line of investigation showed that pancreatic cancer cell lines harbouring KRAS Q61H or ectopically expressing KRAS Q61H were more resistant to SHP2 inhibitor treatment in comparison to cells harbouring other KRAS mutants or WT.

5. In figure 2, how do the authors explain why G13D shows higher intrinsic rate of exchange than WT (panels b, d), and why Q61L is distinct from Q61H (panels b, d)? In panel e, KRAS Q61L should be included for control.

Response:

The higher intrinsic rate of nucleotide exchange of the KRAS G13D mutant has been attributed to the introduction of a negatively charged Asp residue in the nucleotide-binding site near the alpha-phosphate group of GDP/GTP. The Asp side chain creates a repulsive electrostatic force that lowers nucleotide affinity thereby promoting nucleotide exchange. (PMID: 26958611). Replacement of Gln61 with leucine or histidine does not have dramatic impacts on intrinsic exchange (Q61L exchanges slightly faster for reasons that are not clear). However the dramatic difference is the complete resistance to GEF activity of SOS induced by the Q61H

mutant. This is explained by the direct interaction between Q61 and SOS T935, which is critical to support binding (New Fig. 2e,f) This is explained in more detail with references in the manuscript (pages 7-9 and 14-15). In panel e (now panel g) the KRAS Q61L data is included.

6. In figure 3, panel g should contain also a catalytic impaired mutant of SHP2 for control.

Response:

We have shown previously that SHP2 promotes dephosphorylation of Src-induced KRAS phosphorylation, and further showed that catalytically-dead SHP2 C459S is unable to perform this function while oncogenic gain-of-function SHP2 E76K enhances reduces Src-mediated phosphorylation of KRAS (Bunda et al., 2015 Nat Commun; Kano et al., 2019 Nat Commun). We agree, however, with the Reviewer that we should also show in this manuscript the specificity of SHP2. As expected, we show that SHP2 C459S has a diminished ability to reduce the Src-mediated phosphorylation of KRAS in comparison to SHP2 WT (Supplementary Figure S11).

7a. In Figure 5, the model is not clear and contains contradictions. For instance, the authors do not take into account the stoichiometry of SRC and SHP2 activities on WT versus mutant KRAS 61H. Furthermore, is there a quantitative difference in pERK output depending upon the balance between SHP2 and SRC activities?

Response:

We thank the Reviewer for raising this very good point. While we agree that the stoichiometry of SRC and SHP2 activities are an important factor in the KRAS phosphorylation event, it is extremely difficult to measure this at any point of cellular processes. In this model shown in Fig 5, we attempted to illustrate the basic principle and essential components or “players” in this process in order to guide the readers. The point is KRAS function is regulated by not only the commonly accepted GTPase cycle (which involves GEF and GAP regulates), but also the tyrosine phosphorylation event (which is controlled by both SRC kinase and SHP2 phosphatase). In our view, those two regulatory processes play crucial roles, and we hope to highlight the significance of the SRC/SHP2 process by using this simple diagram. We hope that the Reviewer and Editor agree with us on the inclusion of this figure in our paper.

7b. Finally, RAF is referred to generically yet the authors test BRAF RBD more specifically. Why do the authors focus on BRAF RBD and not the more commonly used probe of cRAF RBD and do the findings generalize to all RAF isoforms?

Response:

We thank the Reviewer for pointing this out, and we have now specified which isoform of RAF was used in each experiment. The RBD pulldown experiments were conducted using RAF1 RBD, whereas the BLI experiments use BRAF-RBD. Our findings are consistent for both isoforms, and we expect they would generalize to all RAF isoforms, including ARAF (which was not tested here). All three RAF RBDs bind in the same structural mode and the contacting

residues are highly conserved (PMID: 26165597) In the discussion we refer to RAF generically where we intend to imply all RAF isoforms (page 15).

8. The therapeutic impact of the model is not experimentally validated. What is the most appropriate approach to restoring sensitivity of KRAS G61H to upstream pathway blockade (via SHP2 inhibition) and inhibit cancer cell signaling and viability? This needs to be experimentally shown.

Response:

We thank the Reviewer for this comment, which is important and certainly clinically relevant. Our model is based on our findings that KRAS Q61H is isolated from upstream regulation, thus the model suggests that trying to restore sensitivity to upstream pathway blockade via SHP2 inhibitors is probably not an appropriate therapeutic approach for this particular mutant. Rather, our model points to blockade of the signaling pathways downstream of KRAS Q61H as a more viable option. Therefore, we have performed preliminary experiments to address the Reviewer's comment. To query the effectiveness of targeting downstream components of the RAS signalling pathway as an alternative strategy to target KRAS Q61H, PDAC cells harboring KRAS-G12V or -Q61H mutations were treated with the MEK inhibitor trametinib or the ERK inhibitor ulixertinib alone or in combination (Supplementary Figure S17a). Treatments with single agents showed slight growth inhibition, whereas combination treatments significantly reduced cell viability in both KRAS-mutants PDAC cell lines, compared to vehicle control. Similar results were observed in HEK293T cells engineered to overexpress KRAS-WT, -G12V or -Q61H. (Supplementary Figure S17b). Furthermore, patient-derived organoids bearing KRAS mutations Q61H and G12V exhibited sensitivity to Trametinib and the AKT inhibitor MK-2206 (Supplementary Figure S17c,d) These results suggest that combined downstream MAPK inhibition can be used effectively against KRAS-Q61H pancreatic cancer cells that are resistant to SHP2 inhibition. These preliminary findings support our original speculative statement - 'The present findings suggest that the targeted inhibition of downstream effectors (e.g., RAF, MEK, ERK, PI3K, AKT, mTOR) or combinations thereof may be a more rational and effective approach for the treatment of cancers driven by KRAS Q61H.

RESPONSE TO REVIEWER #3:

The team of Marshall, Ohh and Ikura present a study of the oncogenic Q61H mutant of K-Ras4B. The work provide evidence that SHP2 inhibition have no substantial effect on inhibiting the function of this Ras mutant downstream and furthermore the authors demonstrate this Ras mutant no longer interacts with a key exchange factor protein, SOS1 or GAP protein p120.

Overall this study is well executed and within the ground covered by the authors quite complete. However, I would strongly encourage to expand their study to the molecular / structural level by carrying out now relatively standard model-building for Ras Q61H : SOS1, : p120 and Raf-RBD complexes and perhaps a reasonable all-atom molecular dynamics (~ 300 ns) carried in quadruplicate to equilibrate these models. (the crystal structure of the Q61H is recently become available; one in 2017, one in 2019). In other words what is shown as Fig. 5c should be significantly expanded to the interactions with p120 and Raf-RBD Models with the RBD should probably be run with and without Ras phosphorylation. Only if such data re presented, I would find the study compelling and complete at the molecular level.

Response:

Following the Reviewer's recommendation, we conducted molecular dynamics simulation for KRASQ61H with p120, SOS1, and RBD. Each system was simulated for a total of 900 ns. The requested data with detailed analysis are included in the revised manuscript (New Figures 2e, f, and h; Figure 5c; and Supp Figures 5-9, and 14-16). The results indicate that the KRAS Q61H mutation is disruptive to interactions with p120 GAP and SOS, consistent with our observation that it is resistant to these regulators. Phosphorylation of Tyr32 and Tyr64 increases the dynamics of switch I and II regions in the wild type KRAS and stabilizes the Q61H mutant.

Minor concerns:

1. The authors should mention more prominently that Sos1 and p120 are just two of several regulatory proteins that may act on Q61H in cancer cells. Similarly concerning Raf as one of the effector proteins.

Response:

We thank the Reviewer for this suggestion, and have added a statement that notify the readers as there are other GAPs and GEFs as well as other potential regulatory proteins that act on KRAS. The catalytic domains of all RAS GAPs and GEFs are structurally and mechanistically similar, so we anticipate the impact of the Q61 mutation would be similar on all of them, although this remains unknown. It is possible that the mutation could affect other regulators of KRAS function (e.g., known and novel regulators of localization, multimerization or post-translational modification). As discussed above (R2-Q7b), we expect that the effects of Q61H mutation on all RAF isoforms will be similar, but other effector proteins, particularly those that interact with Switch II (e.g., PI3-kinase, Nore 1 and AGO2) may be impacted in different ways. These ideas have been incorporated into the Discussion (pages 14 and 15).

2. The work of Cruz-Migoni, A et al., PNAS 2019 and Gupta, AK et al., Chem.Biol Drug Des 2019 should be cited.

Response:

These references have been added to the revised manuscript.

3. An overview figure how SHP2 relates to Sos1 and p120 as well as MAPK signaling should be given at the beginning of the paper.

Response:

We thank the Reviewer for this comment; we have included this in the introduction and added a schematic figure showing the interplay of SHP2 with SOS and p120 GAP (Figure 1a).

4. While the real time NMR assays are a critical strength of the paper, it would be good to remind the reader why the curves do not go to zero (e.g. Fig. 2a,c) and different mutants have different off-sets. Also why GTPgammaS is used, rather than GMPPNP in these assays.

Response:

We thank the Reviewer for appreciating the real-time NMR assays. Indeed, it is interesting that the curves do not go to zero and each mutant has a distinct characteristic plateau. This is a result that is not observable in fluorescence-based assays. The curves do not go to zero because the nucleotide exchange equilibrates at a certain ratio, which is a function of the relative affinity of GDP:GTP(analog) and the ratio of the two nucleotides in the sample (1:10 in these experiments to mimic the cellular ratio). We are currently investigating this in more detail, and have added a comment to explain this in Figure 2 legend. We used GTPγS for our NMR study primarily because GTPγS-loaded protein produces better spectra than GMPPNP-loaded KRAS, thus the probe peaks can be monitored more accurately, especially at intermediate time

points when the system comprises a mixture of active and inactive protein, and peaks from each are weaker. Based on the NMR spectra of KRAS, GTP γ S behaves more like native GTP than GMPPNP.

REVIEWER COMMENTS

Reviewer #1 (Remarks to the Author):

The authors have adequately addressed my comments. Congratulations on your excellent work.

Reviewer #2 (Remarks to the Author):

The authors have submitted a revised manuscript to address the comments of each reviewer. While some of my comments were addressed by the authors and this is appreciated, several were not and speculation or correlative observations were instead cited or controls simply not included. While there is value in this interesting work and potential for high impact, I nevertheless cannot yet support publication of the revised manuscript because of continued deficiencies.

Reviewer #3 (Remarks to the Author):

the paper has been adequately revised to address my comments and to my reading also the points made by the other reviewers.

Reviewer #4 (Remarks to the Author):

The search for effective therapies for KRAS mutant cancer is an important, and daunting, field of cancer research. This manuscript builds on previous reports published by the same group that SHP2 can directly dephosphorylate WT and mutant KRAS at Tyr32 and Tyr64 (Src-mediated phosphorylation) promoting its activation. There is some concern regarding novelty, as the resistance to SHP2 inhibitors in different RAS-Q61X isoforms has been reported, along with a number of recent papers investigating these mechanisms of resistance which to the author's credit, are all discussed except for Valencia-Sama et al. CR 2020. Here, the authors report that like WT-KRAS, Tyr32 and Tyr64 of KRAS Q61H can be phosphorylated by Src and dephosphorylated by SHP2. However, none of these processes affects KRAS Q61H output, adding marginal advances to previous reports that linked the resistance to RTK and SHP2 inhibitors to the poor ability of mutant RAS Q61X to "cycle". Another concern is the fact that the SHP2-mediated dephosphorylation of KRAS was mainly studied in vitro or using overexpression systems questioning about the physiological relevance of this finding. Since the first paper published by the same group in 2015, SHP2-mediated dephosphorylation of KRAS at Tyr32 and Tyr64 has not been reproduced yet. Of note, two unbiased phosphoproteomic screens did not identify any changes in the abundance of RAS pTyr sites in cells that were previously activated with RTK ligands (EGF or PDGF) in the presence of the SHP2 inhibitor SHP099 (Batth et al., Cell Rep. 2018; Vemulapalli et al., Elife 2021). New results in figure 3i, added in response to reviewer 2 (#1 question), did not clearly show detectable level of pTyr sites in Hs766T cells (KRAS Q61H). Furthermore, since the blot were cropped, the level of p-RAS in WT and SHP2KO cells treated with or without EGF cannot be fairly compared. In addition, it is not clear if the author used a pan RAS or a KRAS ab for IB after IP with anti -pTyr IP. The manuscript is technically and methodologically sound, and clearly written. However, it seems to me that the above issues really diminish the enthusiasm and the findings remain preliminary. I think that this work may be best suited for a more specialized journal.

REVIEWER COMMENTS

Reviewer #1 (Remarks to the Author):

The authors have adequately addressed my comments. Congratulations on your excellent work.

We thank Reviewer 1 for the positive evaluation of our revised manuscript.

Reviewer #2 (Remarks to the Author):

The authors have submitted a revised manuscript to address the comments of each reviewer. While some of my comments were addressed by the authors and this is appreciated, several were not and speculation or correlative observations were instead cited or controls simply not included. While there is value in this interesting work and potential for high impact, I nevertheless cannot yet support publication of the revised manuscript because of continued deficiencies.

We made a considerable effort to address most of the specific concerns in Reviewer 2's report. We feel that some of the requests were beyond the scope of the paper, and that some of the comments did not provide specific addressable concerns. Similarly, this general evaluation of our revision does not provide concrete concerns that we can either address or dispute.

Reviewer #3 (Remarks to the Author):

the paper has been adequately revised to address my comments and to my reading also the points made by the other reviewers.

We thank Reviewer 3 for the positive evaluation of the revision.

Reviewer #4 (Remarks to the Author):

The search for effective therapies for KRAS mutant cancer is an important, and daunting, field of cancer research, This manuscript builds on previous reports published by the same group that SHP2 can directly dephosphorylate WT and mutant KRAS at tyr32 and tyr64 (Src-mediated phosphorylation) promoting its activation.

There is some concern regarding novelty, as the resistance to SHP2 inhibitors in different RAS-Q61X isoforms has been reported, along with a number of recent papers investigating these mechanisms of resistance which to the author's credit, are all discussed except for Valencia-Sama et al. CR 2020.

We have added this citation to the revised manuscript

Here, the authors report that like WT-KRAS, Tyr32 and Tyr64 of KRAS Q61H can be phosphorylated by Src and dephosphorylated by SHP2. However, none of these processes affects KRAS Q61H output, adding marginal advances to previous reports that linked the resistance to RTK and SHP2 inhibitors to the poor ability of mutant RAS Q61X to "cycle".

We show that Q61H and Q61L have different cycling properties, the biggest difference being that Q61L responds to SOS whereas Q61H is insensitive. We have performed very detailed biochemical and biophysical analyses of the KRAS Q61H mutant and its phosphorylated form, and we like to stress that the new findings and our mechanistic insights into the unique properties of this mutant are highly significant. Previous papers, which focus on the role of SHP2 as an adaptor protein for SOS, did not

provide this level of mechanistic insight into understanding why the Q61H mutation confers resistance to Shp2 inhibitors or the role of SHP2 catalytic function.

Another concern is the fact that the SHP2-mediated dephosphorylation of KRAS was mainly studied in vitro or using overexpression systems questioning about the physiological relevance of this finding. Since the first paper published by the same group in 2015, SHP2-mediated dephosphorylation of KRAS at Tyr32 and Tyr64 has not been reproduced yet. Of note, two unbiased phosphoproteomic screens did not identify any changes in the abundance of RAS pTyr sites in cells that were previously activated with RTK ligands (EGF or PDGF) in the presence of the SHP2 inhibitor SHP099 (Bath et al., Cell Rep. 2018; Vemulapalli et al., Elife 2021).

In this revised manuscript we now have provided more convincing Western blots demonstrating that endogenous KRAS can be tyrosyl phosphorylated and that tyrosyl phosphorylation increases when Shp2 is knocked out, which we will summarize below:

Firstly, earlier this year, a collaborative group involving several institutes in Naples and Perugia, Italy independently reproduced our findings that SHP2 enhances RAS activation by dephosphorylating its inhibitory Tyr32 to trigger the MAPK pathway in thyroid cancer cells (Liotti et al. J Exp Clin Cancer Res 2021).

Secondly, we carefully reviewed the data depositions associated with the unbiased phosphoproteomic screens reported by Bath et al. and Vemulapalli et al. These studies do not report the detection of RAS pTyr at all, thus these papers do not support a notion that the abundance of RAS pTyr does not change in cells treated with a SHP2 inhibitor. While the limitations of RAS pTyr detection in these high-throughput global phosphoproteomic screens are difficult to ascertain, the temporal pattern of pTyr-RAS must be carefully considered following RTK activation in the presence of SHP099 or other SHP2 inhibitors, which can affect the duration and amplitude of phosphorylation of an otherwise rapidly cycling pattern of pTyr-RAS. As discussed further below, the detection of endogenous pTyr-RAS remains technically challenging, however it has been reproduced many times.

Thirdly, there have been several and growing number of reports supporting the notion that RAS is tyrosyl-phosphorylated with potential biological consequences. For example, Fujita-Yamaguchi et al. (PNAS 1989) using purified insulin receptor have shown that RAS can be phosphorylated on Tyr residues, and Uezu et al. (PNAS 2012) identified Tyr32 and Tyr64 as potential sites of phosphorylation on HRAS using a modified SH2 domain to phototrap and identify pTyr proteins from subcellular sites within cells. Zou et al. (JBC 2002) showed that activated Src phosphorylates R-RAS to suppress intracellular integrin activity while Ting et al. (FESEB J 2015) showed that phosphorylation of HRAS on Tyr137 by Abelson tyrosine protein kinase allosterically enhances effector RAF1 binding. As the reviewer indicated, we recently showed tyrosyl-phosphorylation of endogenous wild-type RAS upon PDGF stimulation of mouse embryonic fibroblasts (MEFs), which was markedly diminished in Src/Fyn/Yes knockout MEFs and notably enhanced in CRISPR-mediated SHP2^{-/-} MEFs (Kano et al., Nat Commun 2019). Furthermore, phosphorylation of KRAS Tyr32 and Tyr64 have been detected in a number of other unbiased phosphoproteomic studies, which are catalogued in the PhosphoSitePlus database <https://www.phosphosite.org/siteAction.action?id=15228780> <https://www.phosphosite.org/siteAction.action?id=5840744>.

New results in figure 3i, added in response to reviewer 2 (#1 question), did not clearly show detectable level of pTyr sites in Hs766T cells (KRAS Q61H). Furthermore, since the blot were cropped, the level of

p-RAS in WT and SHP2KO cells treated with or without EGF cannot be fairly compared. In addition, it is not clear if the author used a pan RAS or a KRAS ab for IB after IP with anti -pTyr IP.

The detection of endogenous pTyr-RAS is technically very challenging, due in part to the lack of a high-quality isoform-specific anti-KRAS antibody, thus we rigorously optimized the protocol in the revised manuscript. By blotting Hs766T lysates alongside those of HEK293T cells overexpressing HA-tagged K-, H- and N-RAS using RAS isoform-specific, panRAS and anti-HA antibodies, we determined that the Hs766T cell line expresses K-RAS much more highly than H- or N-RAS (**Supplementary Figure S13**). While the anti-HA blot demonstrates that all three tagged isoforms expressed well, H- and N-RAS were better detected by their respective isoform-specific antibodies than was KRAS (antibody OP24). Despite the differential sensitivities of the three isoform-specific antibodies, the KRAS band was still the main isoform detected in Hs766T. Having established that KRAS is the major RAS isoform in these cells, we used the superior anti-panRAS antibody for immunoprecipitation of KRAS and visualized tyrosyl phosphorylation using a pan-pTyr antibody (4G10). A band signal corresponding to tyrosyl phosphorylated RAS (primarily KRAS) was detected in cells stimulated with EGF, but was negligible in non-stimulated cells (**revised Fig. 3i**).

To compare the levels of pTyr-(K)RAS in WT and SHP2 KO Hs766T, we compared them on the same blot in the revision. We show that p-RAS can be detected in the presence or absence of EGF stimulation when SHP2 is knocked out, which is consistent with the notion that SHP2 dephosphorylates pTyr-RAS. (revised Fig. 3i). We believe that these new data not only address the reviewer's points but also improve and strengthen our findings presented in this paper. We thank this reviewer for the critical and constructive comments.

The manuscript is technically and methodologically sound, and clearly written. However, it seems to me that the above issues really diminish the enthusiasm, and the findings remain preliminary. I think that this work may be best suited for a more specialized journal.

We hope the Reviewer will agree that we have addressed the above issues that diminished enthusiasm and consider the revision suitable for Nature Communications.

REVIEWERS' COMMENTS

Reviewer #4 (Remarks to the Author):

The reviewer appreciated the considerable effort to address all the previous concerns. However, in my opinion, the concerns remain.

1) The authors performed detailed in vitro analyses showing that KRAS Q61H have different cycling properties than Q61L. In particular, Q61L responds to the upstream GEF activity of SHP2/SOS whereas Q61H is insensitive. The authors conclude that this different propriety between Q61L and Q61H confers resistance to SHP2 inhibitors in KRAS Q61H models decoupling this specific mutant from upstream regulation and promoting a "constitutive" GTP ON state of KRAS with downstream activation of the MAPK pathway.

If the model proposed by the author is correct it is not clear why all the KRAS Q61 codon mutants including K/NRAS Q61L models are insensitive to either SHP2 or SOS inhibitors (see Nichols, R. J. et al. Nat Cell Biol 2018; Ahmed, T. A. et al. Cell Rep 2019; Hofmann MH, et al. in Cancer Discovery 2020).

These reports suggested that all the K/NRAS Q61 codon mutants are functionally disconnected from their upstream GEF activity (not only Q61H) because described as "GAP dead" with none intrinsic or GAP mediated (e.g. NF1, RASA1 ect.) activity. Of note, in all the current clinical trials with SHP2 and SOS inhibitors all patients carrying any N/Kras Q61 codon mutant (including Q61L) are excluded. The authors did not performed a direct comparison of the effect of SHP2 inhibition /KO in Q61L vs Q61H using in vivo models (e.g. cell lines, mouse xenografts) to study the differential signaling (RAS-GTP, pMEK, PERK, DUSP6, PRSK3 etc) and/or differential efficacy (cell viability, in vivo tumor growth) of upstream GEF inhibition. I am still of the opinion that this work is only adding marginal advances to previous reports that linked the resistance to RTK, SHP2 and SOS inhibitors to the poor ability of all N/KRAS Q61X to "cycle".

2) I am not still convinced that pTyr KRAS is a substrate of SHP2 in physiological conditions, Revised Fig3 i, in my opinion, does not show clear results of the increase of pTyr KRAS upon SHP2 KO.

I remain of the opinion that all the above concerns were not fully addressed and I still think that this work may be best suited for a more specialized journal.

Reviewer #4 (Remarks to the Author):

The reviewer appreciated the considerable effort to address all the previous concerns.

We thank the reviewer for acknowledging the efforts we have made to address the previous concerns.

However, in my opinion, the concerns remain.

1) The authors performed detailed in vitro analyses showing that KRAS Q61H have different cycling properties than Q61L. In particular, Q61L responds to the upstream GEF activity of SHP2/SOS whereas Q61H is insensitive. The authors conclude that this different propriety between Q61L and Q61H confers resistance to SHP2 inhibitors in KRAS Q61H models decoupling this specific mutant from upstream regulation and promoting a “constitutive” GTP ON state of KRAS with downstream activation of the MAPK pathway.

This summary of the model is correct, although it leaves out the important component that RAF binding upon Tyr phosphorylation is less impaired for the Q61H mutant than for WT KRAS or other mutants tested.

If the model proposed by the author is correct it is not clear why all the KRAS Q61 codon mutants including K/NRAS Q61L models are insensitive to either SHP2 or SOS inhibitors (see Nichols, R. J. et al. Nat Cell Biol 2018; Ahmed, T. A. et al. Cell Rep 2019; Hofmann MH, et al. in Cancer Discovery 2020). These reports suggested that all the K/NRAS Q61 codon mutants are functionally disconnected from their upstream GEF activity (not only Q61H) because described as “GAP dead” with none intrinsic or GAP mediated (e.g. NF1, RASA1 ect.) activity.

Our work showed that two codon 61 mutants (Q61H and Q61L) have very different responses to SOS1 stimulation. KRAS Q61L has fast intrinsic exchange and responds well to SOS1 stimulation of nucleotide exchange. Thus, it effectively differs from the Q61H mutant, which exhibits a slower rate of intrinsic exchange and resists SOS1 stimulation. Here we challenged a general assumption that all Q61X mutants would have similar nucleotide exchange properties (Ahmed, T. A. et al. Cell Rep 2019).

Hofmann MH, et al. (Cancer Discovery 2020) concluded that KRAS Q61H cells are resistant to SOS inhibition, however, the mechanistic speculation focused on this mutant’s impaired GTPase activity but did not address sensitivity to SOS. Indeed, our work showed that KRAS Q61H is resistant to SOS stimulation. We believe our work demonstrates the importance of appreciating multiple biochemical properties of specific KRAS mutations.

Of note, in all the current clinical trials with SHP2 and SOS inhibitors all patients carrying any N/Kras Q61 codon mutant (including Q61L) are excluded.

We thank the Reviewer for bringing this to our attention. We have added two sentences to the paper about how current clinical trials are using RAS mutation status in their eligibility criteria.

The authors did not performed a direct comparison of the effect of SHP2 inhibition /KO in Q61L vs Q61H using in vivo models (e.g. cell lines, mouse xenografts) to study the differential signaling (RAS-GTP, pMEK, PERK, DUSP6, PRSK3 etc) and/or differential efficacy (cell viability, in vivo tumor growth) of upstream GEF inhibition.

The Q61L mutation is common in HRAS, the isoform that was historically better studied for various reasons before it was appreciated that KRAS is much more frequently mutated in human cancers. For historical reasons, Q61L has been used as a 'tool' mutation. In KRAS, the most frequent mutation of codon 61 is Q61H, which is 4.4-fold more frequent than Q61L in the current Cosmic database (742 reports of Q61H vs 169 cases of Q61L, [KRAS ENST00000311936 Gene - Somatic Mutations in Cancer \(sanger.ac.uk\)](https://cancer.sanger.ac.uk/cosmic/gene/analysis/KRAS)). We did not perform this comparison mainly because Q61L mutation is rare in KRAS.

I am still of the opinion that this work is only adding marginal advances to previous reports that linked the resistance to RTK, SHP2 and SOS inhibitors to the poor ability of all N/KRAS Q61X to "cycle".

KRAS Q61H is unique in that its impairment of 'cycling' is more profound than Q61L. In addition, the Q61H mutant is less impaired for RAF binding upon Tyr phosphorylation relative to WT or other mutants tested. Our detailed biochemical and biophysical analyses of the KRAS Q61H mutant and its phosphorylated form reveal important mechanistic insights into the unique properties of this mutant and contribute to understanding why the Q61H mutation confers resistance to Shp2 inhibitors. Importantly, by presenting detailed investigations of KRAS Q61H using computational, biophysical, biochemical and cell biology methods; we demonstrate that not all KRAS Q61X mutants have the same mechanistic phenotypes.

2) I am not still convinced that pTyr KRAS is a substrate of SHP2 in physiological conditions, Revised Fig3 i, in my opinion, does not show clear results of the increase of pTyr KRAS upon SHP2 KO.

In the absence of EGF stimulation, a band corresponding to pTyr-(K)RAS was detected in SHP2 KO (lane 3) but not WT (lane 1) Hs766T cells. We acknowledge this band is diffuse, and this partly because pKRAS is heterogenous, i.e., our in vitro characterization showed that pKRAS exists as a mixture of singly and doubly phosphorylated protein, which migrate differently on SDS-PAGE.

I remain of the opinion that all the above concerns were not fully addressed and I still think that this work may be best suited for a more specialized journal.

We strongly believe that the present work significantly enhances the current mechanistic knowledge of the relationship between the RAS/MAPK pathway and SHP2 inhibition.